# A diffusion-based kernel density estimator (diffKDE, version 1) with optimal bandwidth approximation for the analysis of data in geoscience and ecological research

Maria-Theresia Pelz[1,2], Markus Schartau[2], Christopher J. Somes[2], Vanessa Lampe[2], and Thomas Slawig[1]

[1]Department of Computer Science, Kiel University, 24118 Kiel, Germany
[2]GEOMAR Helmholtz Centre for Ocean Research Kiel, 24105 Kiel, Germany

**Correspondence:** thomas.slawig@email.uni-kiel.de, mschartau@geomar.de

**Abstract.** Probability density functions (PDFs) provide information about the probability of a random variable taking on a specific value. In geoscience, data distributions are often expressed by a parametric estimation of their PDF, such as e.g. a Gaussian distribution. At present there is a growing attention towards the analysis of non-parametric estimation of PDFs, where no prior assumptions about the type of PDF are required. A common tool for such non-parametric estimation is a kernel density estimator (KDE). Existing KDEs are valuable but problematic because of the difficulty of objectively specifying optimal bandwidths for the individual kernels. In this study, we designed and developed a new implementation of a diffusion-based KDE as an open source Python tool to make diffusion-based KDE accessible for general use. Our new diffusion-based KDE provides (1) consistency at the boundaries, (2) better resolution of multimodal data, (3) and a family of KDEs with different smoothing intensities. We demonstrate our tool on artificial data with multiple and boundary close modes and on real marine biogeochemical data, and compare our results against other popular KDE methods. We also provide an example for how our approach can be efficiently utilized for the derivation of plankton size spectra in ecological research. Our estimator is able to detect relevant multiple modes and it resolves modes that are located closely to a boundary of the observed data interval. Furthermore, our approach produces a smooth graph that is robust to noise and outliers. The convergence rate is comparable to the Gaussian estimator, but with a generally smaller error. This is most notable for small data sets with up to around 5000 data points. We discuss the general applicability and advantages of such KDEs for data-model comparison in geoscience.

## 1 Introduction

In geoscience, the application of numerical models has become an integral part of research. Given the complexity of some models, such as Earth system models with their descriptions of detailed processes in the ocean, atmosphere, and land, a number of plausible model solutions may exist. Accordingly, there is a strong demand for the analysis of model simulations on various temporal and spatial scales and to evaluate these results against observational data. A viable evaluation procedure is to compare nonparametric probability density functions (PDFs) of the data with their simulated counterparts. By nonparametric PDFs, it is meant that no assumptions are made regarding any particular (parametric) probability distribution, such as the normal distribution.

Some studies have already documented the advantage of analyzing changes in PDFs, for example, when results of climate models are evaluated on regional scale and their sensitivities to uncertainties in model parameterizations and forcing are examined (e.g. Dessai et al., 2005; Perkins et al., 2007). Likewise, problems of model parameterizations can be approached in a stochastic rather than deterministic framework, which requires a simulated probability distribution to compare well with the probability distribution of truth (Palmer, 2012). Perkins et al. (2007) stressed that the credibility of projecting future distributions of temperature and precipitation is more likely to be good in cases when the PDFs of hindcast simulation results are similar to the PDFs of the observations. A critical point is the quantification of the similarity between respective nonparametric PDFs, typically expressed by some distance or divergence measure, as analyzed and discussed in Thorarinsdottir et al. (2013).

The examination of the suitability of certain divergence functions for data-model assessment is only one aspect. Another is the quality or representativeness of the estimated PDFs. Well approximated PDFs have been used to benefit data analysis in geosciences (e.g. O'Brien et al., 2019; Teshome and Zhang, 2019; Ongoma et al., 2017). Obtaining high quality approximations of nonparametric PDFs is certainly not limited to applications in the geosciences but is likely desirable in other scientific fields as well. In aquatic ecological research, for example, continuous plankton size spectra can be well derived from PDFs of cell size measurements sorted by individual species or plankton groups (e.g. Quintana et al., 2008; Schartau et al., 2010; Lampe et al., 2021). The identification of structural details in the size spectra, such as distinct elevations (modes) and troughs within certain size ranges, is useful, since they can reveal some of the underlying structure of the plankton foodweb. A typical limitation of the approach described in Schartau et al. (2010) and Lampe et al. (2021) is the specification of an estimator for the continuous size spectra, such that all significant details are well resolved.

Mathematically formulated, PDFs are integrable non-negative functions $f : \Omega \to [0, \infty)$ from a sample space $\Omega \subseteq \mathbb{R}$ into the non-negative real numbers with $\int_{-\infty}^{\infty} f(x)\, dx = 1$. PDFs correspond to the probability $P$ of the occurrence of a data value $X \in \mathbb{R}$ within a specific range $[a, b] \subseteq \mathbb{R}$ via the relationship

$$P(a < X < b) = \int_a^b f(x)\, dx \text{ for all } a < b \in \mathbb{R}. \tag{1}$$

The application of kernel density estimators (KDEs) has become a common approach for approximating PDFs in a *nonparametric* way (Parzen, 1962), which means that probability parameters (e.g. expectation or variance) of the data and the type of the underlying probability distribution (e.g., normal or log-normal) are not prescribed. The general concept of KDEs takes into account information of every single data point and treats all of them equally. Consequently, every point's information weighs the same in the resulting estimate without introducing additional assumptions.

A KDE is based on a kernel function and a smoothing parameter. The kernel function is ideally chosen to be a PDF itself, usually unimodal and centered around zero (Sheather, 2004). The estimation process sums up the kernel function sequentially centered around each data point. The sum of these individual kernels is standardized by the number of data points. This ensures that the final estimate is again a PDF by inheriting all properties of its kernels. The smoothing parameter, referred to as bandwidth, determines the smoothness of the estimate. If it is chosen to be small, more details of the underlying data structure become visible. If it is larger, more structure becomes smoothed out (Jones et al., 1996), and information from

single data points can get lost. Hence, it is crucial to determine some kind of an optimal size of the bandwidth parameter to represent a suitable signal-to-noise ratio that allows a separation of significant distinctive features from ambiguous details. The question of optimal bandwidth selection is widely discussed in the literature (e.g., Sheather and Jones, 1991; Jones et al., 1996; Heidenreich et al., 2013; Chacón and Duong, 2018; Gramacki, 2018). It also takes into account that there might not be one single "optimal" choice for such bandwidth (Abramson, 1982; Terrell and Scott, 1992; Chaudhuri and Marron, 2000; Scott, 2012), to use adaptive bandwidth approaches (Davies and Baddeley, 2017) or to optimize the kernel function shape instead of the bandwidth (Bernacchia and Pigolotti, 2011).

The reformulation of the most common Gaussian KDEs (Sheather, 2004) into a diffusion equation provides a different view on KDE (Chaudhuri and Marron, 2000). This different approach comes with three main advantages: (1) consistency at the boundaries (2) better resolution of multimodal data (3) a family of KDEs with different smoothing intensities can be produced as a by-product of the numerical solution. This perspective change is possible because the Gaussian kernel function solves the partial differential equation describing the diffusion heat process as the Green function. The time parameter of this differential equation corresponds to the smoothness of the estimate, and thus becomes tantamount to the estimate's bandwidth parameter (Chaudhuri and Marron, 2000). The initial value is typically set to include the $\delta$-distribution of the input data, which will formally be defined in Sec. 3. This differentiates the initial value problem from classical problems, since the $\delta$-distribution is not a proper function itself. In specific applications this diffusion approach delivered convincing results (e.g., Botev et al., 2010; Deniz et al., 2011; Qin and Xiao, 2018), especially for the resolution of multiple modes (e.g. Majdara and Nooshabadi, 2020). The improved structure resolution has for example already shown useful for the optimization of photovoltaic power generation (Li et al., 2019), analysis of flood frequencies (Santhosh and Srinivas, 2013) or the prediction of wind speed (Xu et al., 2015). However, it tends to resolve too many details or overfit the data in others (e.g., Ma et al., 2019; Chaudhuri and Marron, 2000; Farmer and Jacobs, 2022). One main benefit of the diffusion KDE is that it provides a series of PDF estimates for a sequence of bandwidths by default (Chaudhuri and Marron, 2000). As a consequence, it offers the chance to choose between varying levels of smoothness by design.

In this study, we present a new, modified diffusion-based KDE with an accompanying Python package, diffKDE (Pelz and Slawig, 2023). Our aim is to retain the original idea of diffusion-based KDEs by Chaudhuri and Marron (2000) and Botev et al. (2010), but to avoid the complex fixed-point iteration by Botev et al. (2010). The main objective of our refined approach is to achieve high performance for analyses of high variance and multimodal data sets. Our diffusion-based KDE is based on an iterative approximation that differs from others, using a default optimal bandwidth and two preliminary *pilot* estimates. This way the KDE can provide a family of estimates at different bandwidths to choose from in addition to a default solution optimally designed for data from geoscience and ecological research. This allows for an interactive investigation of estimated densities at different smoothing intensities.

This paper is structured as follows: At first, we will briefly recall the general concept of KDEs. Afterwards, our specific KDE approach will be introduced and described, as developed and implemented in our software package. We explain the two pilot estimation steps and the selection of the smoothing parameters. Then the performance of our refined estimator will be compared with other state-of-the-art KDEs, while considering known distributions and real marine biogeochemical data.

The real test data include carbon isotope ratios of particulate organic matter ($\delta^{13}C_{POC}$) and plankton size data. Our analyses presented here involve investigations of KDE error, runtime, the sensitivity to data noise and the characteristics of convergence w.r.t. increasing sample size.

## 2 The general kernel density estimator

A kernel density estimator (KDE) is a non-parametric statistical tool for the estimation of PDFs. In practice, diverse specifications of KDEs exist that may improve the performance with respect to individual needs. Before we explain our specifications of the diffusion-based KDE, we will provide basic background information about KDEs.

For all following let $\Omega \subseteq \mathbb{R}$ be a domain, $X_j \in \Omega, j \in \{1, ..., N\}$, be $N \in \mathbb{N}$ independent identically distributed real random variables.

The most general form of a KDE approximates the true density $f$ of the input data $(X_j)_{j=1}^N$ by

$$\hat{f}(x; h) = \frac{1}{Nh} \sum_{j=1}^N K\left(\frac{x - X_j}{h}\right), \text{ where } x \in \mathbb{R}, h \in \mathbb{R}_{>0} \text{ and } \hat{f}(x; h) \mathbb{R}_{\geq 0}. \tag{2}$$

The sets $\mathbb{R}_{>0}$ and $\mathbb{R}_{\geq 0}$ denote the positive real numbers and the non-negative real numbers, respectively. The *kernel function* $K : \mathbb{R} \to \mathbb{R}$ satisfies the following conditions (Parzen, 1962):

$$\sup_{y \in \mathbb{R}} |K(y)| < \infty, \quad \int_{\mathbb{R}} |K(y)| dy < \infty, \quad \lim_{y \to \infty} |yK(y)| = 0, \quad \int_{\mathbb{R}} K(y) dy = 1, \tag{3}$$

meaning that $K$ is bounded, integrable, and for the limit $y \to \infty$ decreases faster to zero than $y$ approaches infinity. The final condition means that $K$ integrates to 1 over the whole real domain, which implies that also the KDE $\hat{f}$ integrates to 1 as it is necessary for a PDF.

The parameter $h$ determines the smoothness of the estimate calculated by Eq. 2 and is called the *bandwidth parameter* (Silverman, 1986). An optimal choice for the bandwidth parameter is regarded as the minimizer of the asymptotic mean integrated squared error between the true density of $(X_j)_{j=1}^N$ and their KDE (Sheather and Jones, 1991). The mean integrated squared error (**MISE**) is defined as

$$\text{MISE}\left(\hat{f}\right)(h) = \mathbf{E}\left(\int_{\mathbb{R}} \left(\hat{f}(x; h) - f(x)\right)^2\right) dx, \text{ where } h \in \mathbb{R}_{>0} \tag{4}$$

for all PDFs $f$ and respective KDEs $\hat{f}$ (Scott, 1992). In the following, we will work with the asymptotic **MISE** denoted as **AMISE**, which describes the asymptotic behavior of the **MISE** for the bandwidth parameter approaching zero $h \to 0$, meaning

$$\lim_{h \to 0} \frac{\text{MISE}\left(\hat{f}\right)(h)}{\text{AMISE}\left(\hat{f}\right)(h)} = 1. \tag{5}$$

If now $\hat{f}$ is a KDE and there exists a $h^* \in \mathbb{R}_{>0}$ with

$$\mathbf{AMISE}\left(\hat{f}\right)(h^*) = \min_{h \in \mathbb{R}_{>0}} \mathbf{AMISE}\left(\hat{f}\right)(h), \tag{6}$$

we call $h^*$ the *optimal bandwidth* of $\hat{f}$ by $(X_j)_{j=1}^N$ (Scott, 1992). A kernel function $K$ that suffices the additional conditions

$$\int_{\mathbb{R}} yK(y)\,dy = 0, \int_{\mathbb{R}} y^2 |K(y)|\,dy < \infty, \int_{\mathbb{R}} y^2 K(y)\,dy = k_2 \in \mathbb{R} \setminus \{0\}, \tag{7}$$

is a second order kernel as its second moment $\int_{\mathbb{R}} y^2 K(y)\,dy$ is its first non-zero moment. Those kernels are positive and together with the final condition from Eq. 3 they are PDFs themselves. For the general KDE from Eq. 2 with a second order kernel, the optimal bandwidth can be calculated as

$$h^* = \left(\frac{\|K\|_{L^2}^2}{N k_2^2 \|f''\|_{L^2}^2}\right)^{\frac{1}{5}}. \tag{8}$$

This result was lready obtained by Parzen (1962) and is in more detail discussed in App. A.

In Eq. 8, the true density $f$ is involved in the calculation of the optimal bandwidth $h^*$, which is in turn needed for the approximation of $f$ by a KDE, and thus prevents a direct derivation of an optimal bandwidth. One possibility of how this implicit relation can be solved is the calculation of pilot estimation steps. Our specific approach to this is shown in Sec. 3.1 and Sec. 3.2.

There exists a variety of available choices for the type of kernel function $K$, which all have their individual benefits and shortcomings. Amongst them are, for example, the uniform, triangle, or the Epanechnikov kernel (Scott, 1992):

$$K_E(w) = \frac{3}{4}\left(1 - w^2\right), \text{ where } w \in \mathbb{R} \text{ and } K_E(w) \in \mathbb{R}_{\geq 0}. \tag{9}$$

A common choice for $K$ is the Gaussian kernel (Sheather, 2004):

$$\Phi(w) = \frac{1}{\sqrt{2\pi}} e^{-\frac{1}{2} w^2}, \text{ where } w \in \mathbb{R} \text{ and } \Phi(w) \in \mathbb{R}_{\geq 0}. \tag{10}$$

The standard KDE from Eq. 2 – despite being widely applied and investigated – comes with several disadvantages in practical applications (Khorramdel et al., 2018). For example, severe boundary bias can occur when applied on a compact interval (Marron and Ruppert, 1994). It means that a kernel function with a specified bandwidth, attributed to a single point nearby the boundary, may actually exceed the boundary. Furthermore, it can lack a proper response to variations in the magnitude of the true density $f$ (Breiman et al., 1977). The introduction of a parameter that depends on the respective data region can address the latter (Breiman et al., 1977). Unfortunately, no true independent local bandwidth strategy exists (Terrell and Scott, 1992), meaning that in all local approaches there is still an influence of neighboring data points on each locally chosen bandwidth.

## 3 The diffusion-based kernel density estimator

The diffusion-based KDE provides a different approach to Eq. 2 by solving a partial differential equation instead of the summation of kernel functions. This different calculation offers three main advantages: (1) consistency at the boundaries can be

ensured by adding Neumann boundary conditions, (2) better resolution of multimodal data can be achieved by the inclusion of a suitable parameter function in the differential equation leading to adaptive smoothing intensity, (3) a family of KDEs with different bandwidths is produced as a by-product of the numerical solution of the partial differential equation in individual time steps.

This KDE solves Eq. 11, the partial differential equation describing the diffusion heat process, starting from an initial value based on the input data $(X_j)_{j=1}^N$ and progresses forward in time to a final solution at a fixed time $T \in \mathbb{R}_{>0}$.

$$\frac{\partial}{\partial t} u(x;t) = \frac{1}{2} \frac{d^2}{dx^2} u(x;t), x \in \Omega, t \in \mathbb{R}_{>0} \tag{11}$$

The input data are treated as the initial value $u(x,0)$ at the initial time $t_0 = 0$ and generally set to infinitely high peaks at every data point $X_j, j \in \{1,...,N\}$. The time propagation in solving Eq. 11 smooths the initial shape of $u$ meaning that $u$ contains

less details of the input data $(X_j)_{j=1}^N$ for increasing values in time $t \in \mathbb{R}_{>0}$. If we observe the solution $u$ of Eq. 11 at a specific fixed final iteration time $T \in \mathbb{R}_{>0}$, this parameter determines the smoothness of the function $u$ and how many details of the input data are resolved. This is an equivalent dependency as already seen for the KDE as the solution of Eq. 2 depending on a bandwidth parameter $h \in \mathbb{R}_{>0}$.

    An advantageous connection to Eq. 2 is that the widely applied Gaussian kernel is a fundamental solution of this differential

equation. Precisely, the Gaussian kernel from Eq. 10 as applied in the construction of a Gaussian KDE depends on the location $x \in \mathbb{R}$ and the smoothing parameter $h \in \mathbb{R}_{>0}$ and has the form

$$\Phi(x;t) = \frac{1}{\sqrt{2\pi}} e^{-\frac{1}{2}\left(\frac{x-X_j}{\sqrt{t}}\right)^2}, \text{ where } x \in \mathbb{R}, t \in \mathbb{R}_{>0} \text{ and } \Phi(x;t) \in \mathbb{R}_{\geq 0}. \tag{12}$$

This function solves Eq. 11 as the Green's function, where the time parameter $t \in \mathbb{R}_{>0}$ equals the squared bandwidth parameter $h^2$ (Chaudhuri and Marron, 2000). Consequently, we can use the result of the optimal bandwidth from Eq. 8, only as the squared

result as

$$T = \left(\frac{\|K\|_{L^2}^2}{N k_2^2 \|f''\|_{L^2}^2}\right)^{\frac{2}{5}}, \tag{13}$$

where we denote the optimal bandwidth now with $T \in \mathbb{R}_{>0}$ as this is the final iteration time in the solution of Eq. 11. This idea to use the diffusion heat equation to calculate a KDE was first proposed by Chaudhuri and Marron (2000) and its benefits were widely explored in Botev et al. (2010).

Our implementation of the diffusion KDE is based on Chaudhuri and Marron (2000), which we extended by some advancements proposed by Botev et al. (2010): We included a parameter function $p \in C^2(\Omega, \mathbb{R}_{>0})$ with $\|p''\|_\infty < \infty$ into Eq. 11, acting inversely to a diffusion quotient. This parameter function allows to influence the intensity of the diffusion applied adaptively depending on the location $x \in \mathbb{R}$. Its role and specific choice is discussed in detail in Sec. 4. Boundary conditions are set to be Neumann and the initial value being a normalized sum of the $\delta$-distributions centered around the input data points.

In the following, we call a function $u \in C^{2,1}(\Omega \times \mathbb{R}_{>0}, \mathbb{R}_{\geq 0})$ the *diffusion kernel density estimator* (diffKDE) if it solves the

diffusion partial differential equation

$$\frac{\partial}{\partial t} u(x;t) = \frac{1}{2} \frac{d^2}{dx^2} \left( \frac{u(x;t)}{p(x)} \right), \qquad\qquad x \in \Omega, t \in \mathbb{R}_{>0}, \tag{14}$$

$$\frac{\partial}{\partial x} \left( \frac{u(x;t)}{p(x)} \right) = 0, \qquad\qquad x \in \partial\Omega, t \in \mathbb{R}_{>0}, \tag{15}$$

$$u(x;0) = \frac{1}{N} \sum_{j=1}^{N} \delta(x - X_j), \qquad\qquad x \in \Omega. \tag{16}$$

The final iteration time $T \in \mathbb{R}_{>0}$ of the solution process of Eq. 14 is called the squared *bandwidth* of the diffKDE. In Eq. 16, the data are incorporated as initial values via the Dirac $\delta$-distribution, i.e., a generalized function which takes the value infinity at its argument and zero anywhere else. In general, $\delta$ is defined by $\delta(x) = 0$ for all $x \in \mathbb{R} \setminus \{0\}$ and $\int_{-\infty}^{\infty} x \delta(x) = 1$ (Dirac, 1927). When regarded as a PDE, the Dirac $\delta$-distribution puts all of the probability as the corresponding data point. The $\delta$-distribution can be defined exactly as a limit of functions, the so-called Dirac sequence. In actual implementations, it has to
be approximated, see Sec. 5.3.

This specific type of KDE has several advantages. First of all, it naturally provides a sequence of estimates for different smoothing parameters (Chaudhuri and Marron, 2000). This makes the identification of one single optimal bandwidth unnecessary, which is ideal because the optimal value can be specific to a certain application and is often debated (e.g., Abramson, 1982; Terrell and Scott, 1992; Chaudhuri and Marron, 2000; Scott, 2012). Aditionally, such a sequence allows a specification
of the estimate's smoothness that is most appropriate for the analysis. The parameter function $p$ introduces adaptive smoothing properties (Botev et al., 2010). Thus, choosing $p$ to be a function allows for a spatially dependent influence on the smoothing intensity, which solves the prior problem of having to locally adjust the bandwidth to the respective region to prevent over-smoothing of local data structure (Breiman et al., 1977; Terrell and Scott, 1992; Pedretti and Fernàndez-Garcia, 2013). In contrast to local bandwidth adjustments, local variations of the smoothing intensity can be applied to resolve multimodal data
as well as values close to the boundary.

### 3.1   Bandwidth selection

We now focus on the selection of the optimal squared bandwidth $T \in \mathbb{R}_{>0}$ according to the relationship $T = h^2$ between final iteration time $T$ of the diffKDE and bandwidth parameter $h$ (Chaudhuri and Marron, 2000),. In the following we refer to this as the *bandwidth selection* for simplicity.
We stressed in Eq. 13 that the optimal choice of the bandwidth parameter depends on the true density $f$. In our setup of the diffKDE, the analytical solution for $T$ of Eq. 6 depends not only on the true density $f$, but also on the parameter function $p$. It can be calculated as

$$T^* = \left( \frac{\mathbf{E}\left(\sqrt{p(X)}\right)}{2N\sqrt{\pi} \left\| \left(\frac{f}{p}\right)'' \right\|_{L^2}^2} \right)^{\frac{2}{5}}, \tag{17}$$

where $\|\cdot\|_{L^2}$ is the $L^2$-norm and $\mathbf{E}(\cdot)$ the expected value. The proof of this equation is in detail given in Botev et al. (2010). The role of the parameter function $p$ is in detail described in Sec. 3.2.

In the simplified setup with $p = 1$ as in Eq. 11, the analytical optimal solution of Eq. 13 becomes

$$T^*_{(p=1)} = \left( \frac{1}{2 N \sqrt{\pi} \|f''\|^2_{L^2}} \right)^{\frac{2}{5}}. \tag{18}$$

Still, the smoothing parameter depends on the unknown density function $f$ and its derivatives. So we will need to find a suitable approximation of $f$, which might again be dependent of $f$ and p. This implicit dependency can be solved by so-called pilot estimation steps. Pilot estimates are generally rough estimates of $f$ calculated in an initial step to use them for an approximation of Eq. 17 and 18, which later on serve to calculate a more precise estimate of $f$. A more detailed introduction into pilot estimation and its specific benefit for diffusion-based KDEs is presented in Sec. 3.2.

Botev et al. (2010) used an iterative scheme to solve the implicit dependency of the bandwidth parameter on the true distribution $f$. This additional effort is avoided in our approach by directly approximating $f$ with a simple data-based bandwidth approximation based on two pilot estimation steps in detail described in Sec. 4.

The possible difficulties in finding one single optimal bandwidth (e.g. Scott, 2012) do not arise in the calculation of the diffKDE by default. This problem is solved by creating a family of estimates from different bandwidth parameters (Breiman et al., 1977), ranging from oversmoothed estimates to those with beginning oscillations (Sheather, 2004). For the diffKDE, the progression of the time $t$ up to a final iteration time $T$ is equivalent to the creation of such a family of estimates. We thus only need to find a suitable optimal final iteration time $T^*$. Then the temporal solution of Eq. 14 provides solutions for the diffKDE for the whole sequence of the temporal discretization time steps smaller than $T^*$, which we can use as the requested family of estimates.

## 3.2 Pilot estimation

A crude first estimate of the true density $f$ can serve as a pilot estimation step for several purposes (Abramson, 1982; Sheather, 2004). The most obvious in our case is to obtain an estimate of $f$ for the calculations of the optimal bandwidth in Eq. 17. The second purpose is its usage for the definition of the parameter function $p$ in Eq. 14. Setting this as an estimate of the true density itself introduces locally adaptive smoothing properties (Botev et al., 2010). Since $p$ appears in the denominator in the diffusion equation, it operates conversely to a classical diffusion coefficient. Choosing $p$ to be a function allows for a spatially dependent influence on the smoothing intensity as follows. At points where the function $p$ is small, the smoothing becomes more pronounced. Whereas if $p$ is larger, the smoothing is less intense. Low smoothing resolves more variability within areas with many similar values (high density), while the intensity of smoothing is increased where data values are more dispersed. Eventually, we calculate two pilot estimates – one for $p$ and one for $f$ – to support the calculation of the diffKDE. We set both pilot estimates to be the solution of Eq. 11 with an optimal smoothing parameter approximating Eq. 18. This approach combines Gaussian KDE and diffKDE interchangeably to make best use of both of their benefits (Chung et al., 2018).

## 4 The new bandwidth approximation and pilot estimation approach of the diffKDE

Our new approach solves the diffusion equation in three stages, where the first two provide pilot estimation steps for the diffKDE. The three chosen bandwidths increase in complexity and accuracy over this iteration. This algorithm was implemented in Python 3.

For the optimal final iteration time $T^*$ from Eq. 17, we need the parameter function $p$ as well as the true density $f$. We approximate them both by a simple KDE, each as pilot estimation steps. We use for both cases the simplified diffKDE defined in Eq. 11, without additional parameter functions. We denote the final iteration times for $p$ and $f$ as $T_p, T_f \in \mathbb{R}_{>0}$, respectively. We use a simple bandwidth as variants of the *rule of thumb* by Silverman (1986) to calculate both of them.

We begin to estimate $T_p$, which is the final iteration time for the KDE that serves as $p$. It shall be the smoothest of the three estimates, since $p$ limits the resolution fineness of the diffKDE as a lower boundary. This occurs because the diffKDE converges to this parameter function and hence never resolves less details than $p$ itself (Botev et al., 2010).

As seen in Eq. 18, the optimal bandwidth for the approximation of $p$ is depending on the second derivative of $f$. We therefore need to make some initial assumption about $f$. For a first simplification, we assume that $f$ belongs to the normal distribution family. Then the variance can be estimated by the standard deviation of the data. This leads us to the parametric approximation of the bandwidth $T_P$ (Silverman, 1986)

$$T_p = \left( \frac{1}{2N\sqrt{\pi}\|f''\|_{L^2}^2} \right)^{\frac{2}{5}} = \left( \frac{1}{2N\sqrt{\pi}\sigma^{-5}\|\Phi''\|_{L^2}^2} \right)^{\frac{2}{5}} = \left( \frac{1}{2N\sqrt{\pi}\sigma^{-5}\frac{3}{8}\frac{1}{\sqrt{\pi}}} \right)^{\frac{2}{5}} = \sigma^2 \left( \frac{4}{3}N \right)^{-\frac{2}{5}}, \tag{19}$$

whose estimate is known to be overly smooth on multimodal distributions.

To calculate the final iteration time $T_f$ for the approximation of $f$ in Eq. 17 , we choose a refined approximation of Eq. 18 that has been proposed by Silverman (1986) as

$$T_f = \left( 0.9\,min\left( \sigma, \frac{iqr\,(data)}{1.34} \right) \right)^2 N^{-\frac{2}{5}}. \tag{20}$$

The $iqr$ is the interquartile range defined as $iqr\,(data) = q\,(0.75) - q\,(0.25)$. The value $q\,(0.25)$ denotes the lower quartile and describes the value in $data$, at which 25 % of the elements in $data$ have a value smaller than $q\,(0.25)$. $q\,(0.75)$ denotes the upper quartile and describes the analogue value for 75 % (Dekking et al., 2005).

We approximate optimal final iteration time $T^*$ from Eq. 17 by calculating $p$ and $f$ by Eq. 11, based on Eq. 19, and Eq. 20 respectively, on an equidistant spatial grid $(x_i)_{i=0}^n \subseteq \bar{\Omega}$ of the spatial domain $\Omega \subseteq \mathbb{R}$. The nominator is approximated by the unbiased estimator and denoted as $E_\sigma \in \mathbb{R}$ (Botev et al., 2010)

$$\mathbf{E}\,(p(X)) \approx \frac{1}{n+1} \sum_{i=0}^{n+1} \sqrt{p\,(x_i)} = E_\sigma \tag{21}$$

and the second derivative in the denominator by finite differences (McSwiggan et al., 2016) and set to $q_i \in \mathbb{R}$ for all $i \in \{1,...,n\}$

$$\left( \frac{f}{p} \right)''(x_i) \approx \frac{1}{h^2} \left( \frac{f}{p}\,(x_{i+1}) - 2\frac{f}{p}\,(x_i) + \frac{f}{p}\,(x_{i-1}) \right) = q_i. \tag{22}$$

For the boundary values we set the second derivative at the lower boundary to $q_0 \in \mathbb{R}$

$$\left(\frac{f}{p}\right)''(x_0) \approx \frac{1}{h^2}\left(2\frac{f}{p}(x_1) - 2\frac{f}{p}(x_0)\right) = q_0 \tag{23}$$

and the second derivative at the upper boundary to $q_{n+1} \in \mathbb{R}$

$$\left(\frac{f}{p}\right)''(x_{n+1}) \approx \frac{1}{h^2}\left(2\frac{f}{p}(x_{n-1}) - 2\frac{f}{p}(x_n)\right) = q_{n+1}. \tag{24}$$

We set the finite differences approximation from Eq. 22, Eq. 23 and Eq. 24 as a discrete function with image $\boldsymbol{q} := (q_0, ..., q_{n+1})$.

In this way we derived an already discrete formula for approximation of the optimal squared bandwidth $T^* \in \mathbb{R}_{>0}$ of the diffKDE on the discretization $(x_i)_{i=0}^n$ of $\Omega$ as

$$T^* = \left(\frac{E_\sigma}{2N\sqrt{\pi}\|\boldsymbol{q}\|_{L^2}^2}\right)^{\frac{2}{5}}. \tag{25}$$

The $L^2$-norm is calculated on the discretized versions of $f$ and $p$ by array operations. The integration is performed by the *trapz* function of the SciPy integrate package (Gommers et al., 2022), the square root is part of the math package (Van Rossum, 275 2020).

## 5 Discretization and implementation of the diffKDE

Equation 14 is solved numerically using a spatial and temporal discretization. The discretization is based on finite differences and sparse matrices in Python. A similar approach can be found in a diffusion-based kernel density estimator for linear networks implemented in R by McSwiggan et al. (2016).

### 5.1 Spatial discretization

We start with the description of the discretization of the spatial domain $\Omega \subseteq \mathbb{R}$. This will reduce the partial differential equation in Eq. 14 into a system of linear ordinary differential equations. Let $n \in \mathbb{N}$ and $(x_i)_{i=0}^n \subseteq \bar{\Omega}$, an equidistant discretization of $\Omega$ with $x_{i-1} < x_i$ and spatial discretization step size $\mathbb{R}_{>0} \ni h = x_i - x_{i-1}$ for all $i \in \{1, \ldots, n\}$. For the following calculations, we set $x_{-1} = x_0 - h \in \mathbb{R}$ and $x_{n+1} = x_n + h \in \mathbb{R}$. Let $u$ be the solution of the diffKDE and $p$ its parameter function, both as 285 defined in Sec. 3. We assume that $u$ and $p$ are both defined on $x_{-1}$ and $x_{n+1}$ and we set $u_i = u(x_i)$ and $p_i = p(x_i)$ for all $i \in \{-1, \ldots, n+1\}$.

Let $t \in \mathbb{R}_{>0}$. We approximate Eq. 15 at $x = x_0$ by applying a first order central difference quotient as

$$0 = \frac{\partial}{\partial x}\left(\frac{u(x_0; t)}{p(x_0)}\right) \approx \frac{1}{2h}\left(\frac{u_1(t)}{p_1} - \frac{u_{-1}(t)}{p_{-1}}\right).$$

This implies

$$\frac{u_{-1}(t)}{p_{-1}} = \frac{u_1(t)}{p_1}.$$

We approximate Eq. 14 at $x = x_0$ by applying a second order central difference quotient

$$u_0'(t) \approx \frac{1}{2}\frac{1}{h^2}\left(\frac{u_1(t)}{p_1} - 2\frac{u_0(t)}{p_0} + \frac{u_{-1}(t)}{p_{-1}}\right) = \frac{1}{2}\frac{1}{h^2}\left(2\frac{u_1(t)}{p_1} - 2\frac{u_0(t)}{p_0}\right). \tag{26}$$

Analogously, we approximate Eq. 15 and Eq. 14 at $x = x_n$ again by first and second order central difference quotients, respectively. This gives

$$u_n'(t) \approx \frac{1}{2}\frac{1}{h^2}\left(\frac{u_{n+1}(t)}{p_{n+1}} - 2\frac{u_n(t)}{p_n} + \frac{u_{n-1}(t)}{p_{n-1}}\right) = \frac{1}{2}\frac{1}{h^2}\left(2\frac{u_{n-1}(t)}{p_{n-1}} - 2\frac{u_n(t)}{p_n}\right). \tag{27}$$

Finally, we derive from Eq. 14 by applying a second order central difference quotient for all $i \in \{1, \ldots, n-1\}$:

$$u_i'(t) \approx \frac{1}{2}\frac{1}{h^2}\left(\frac{u_{i+1}(t)}{p_{i+1}} - 2\frac{u_i(t)}{p_i} + \frac{u_{i-1}(t)}{p_{i-1}}\right). \tag{28}$$

Now, we identify $\boldsymbol{p} := (p_0, \ldots, p_n) \in \mathbb{R}^{n+1}$, $\boldsymbol{u}'(t) := (u_0'(t), \ldots, u_n'(t)) \in \mathbb{R}^{n+1}$ and $\boldsymbol{u}(t) := (u_0(t), \ldots, u_n(t)) \in \mathbb{R}^{n+1}$ with their spatial discretizations. Furthermore, we define $\boldsymbol{v}_{\text{upper}} := (2, 1, \ldots, 1) \in \mathbb{R}^n$, $\boldsymbol{v}_{\text{main}} := (-2, \ldots, -2) \in \mathbb{R}^{n+1}$ and $\boldsymbol{v}_{\text{lower}} := (1, \ldots, 1, 2) \in \mathbb{R}^n$ to be the upper, main and lower diagonal of the tridiagonal matrix $\mathbf{V} \in \mathbb{R}^{(n+1) \times (n+1)}$. Now, we set

$$\frac{1}{2}\frac{1}{h^2}\mathbf{V}\frac{1}{\boldsymbol{p}} = \mathbf{A} \in \mathbb{R}^{(n+1) \times (n+1)}, \tag{29}$$

where $\mathbf{A} \in \mathbb{R}^{(n+1) \times (n+1)}$ means that $\mathbf{A}$ has real entries and $n+1$ rows and $n+1$ columns, the division by $\boldsymbol{p}$ is applied column-wise. Then, Eq. 26, Eq. 27 and Eq. 28 can be summarized as a linear system of ordinary differential equations:

$$\boldsymbol{u}'(t) \approx \frac{1}{2}\frac{1}{h^2}\mathbf{V}\frac{\boldsymbol{u}(t)}{\boldsymbol{p}} = \mathbf{A}\boldsymbol{u}(t). \tag{30}$$

By these calculations the solution of the partial differential equation from Eq. 14 can also be approximated by solving the system of ordinary differential equations:

$$\boldsymbol{u}'(t) = \mathbf{A}\boldsymbol{u}(t), \quad t \in \mathbb{R}_{>0}. \tag{31}$$

## 5.2 Temporal discretization

The time-stepping applied to solve the ordinary differential equation from Eq. 31 and Eq. 16 is again built on equidistant steps forward in time. Let $\Delta \in \mathbb{R}_{>0}$ small and set $t_0 = 0$ and $t_k = t_{k-1} + \Delta$ for all $k \in \mathbb{N}$. Set $u_{k,i} = u(x_i, t_k) \in \mathbb{R}$ for all $i \in \{0, \ldots, n+1\}$ and $k \in \mathbb{N}_0$ and identify $\boldsymbol{u_k} = (u_{k,i})_{i=0}^n \in \mathbb{R}^{n+1}$ for all $k \in \mathbb{N}_0$ with their discretizations.

We use an implicit Euler method to approximate Eq. 31 for all $k \in \mathbb{N}_0$

$$\boldsymbol{u_{k+1}} = \Delta\mathbf{A}\boldsymbol{u_{k+1}} + \boldsymbol{u_k} \tag{32}$$

from which we obtain

$$\boldsymbol{u_k} = (\mathbf{I}_{n+1} - \Delta\mathbf{A})\boldsymbol{u_{k+1}} \text{ for all } k \in \mathbb{N}_0. \tag{33}$$

The implicit Euler method is chosen at this place, since it is A-stable and by this ensures convergence of the solver.

Eq. 33 together with the initial value Eq. 16 describes an implementation-ready time stepping procedure. The linear equation for $\boldsymbol{u_{k+1}}$ will be solved in every time step $k \in \mathbb{N}_0$.

## 5.3 Initial value

The initial value in Eq. 16 depends on the $\delta$-distribution (Dirac, 1927). The $\delta$-distribution is not a proper function, but can be calculated as a limit of a suitable function sequence. A common approximation for the $\delta$-distribution is to use a Dirac sequence (Hirsch and Lacombe, 1999). Such is a sequence $(\Phi_n)_{n\in\mathbb{N}}$ of integrable functions that are non-negative and satisfy

$$\int \Phi_n(x)\,dx = 1 \text{ for all } n \in \mathbb{N} \tag{34}$$

and

$$\lim_{n\to\infty} \int_{\mathbb{R}\setminus\mathcal{B}_\rho(0)} \Phi_n\,dx = 0 \text{ for all } \rho \in \mathbb{R}_{>0}, \tag{35}$$

where $\mathcal{B}_\rho(0) = \{x \in \mathbb{R}; |x-0| < \rho\} = (-\rho, \rho)$ is the open subset of R centered around $\mathbb{R}$ with radius $\rho$. For our implementation we define a Dirac sequence $(\Phi_h)_{h\in\mathbb{R}_{>0}}$ depending on the spatial discretization step size $h \in \mathbb{R}_{>0}$ as an approximation of $\delta$ in Eq. 16. The spatial discretization step size $h \in \mathbb{R}_{>0}$ equals the length of the domain $|\Omega|$ divided by the number of spatial discretization points $n$, namely $h = \frac{|\Omega|}{n}$. This relationship provides the dependency of $\Phi_h$ on $n \in \mathbb{N}$ and the equivalence of the limits $n \to \infty$ and $h \to 0$ in this framework. In the following we give the specific definition of our function sequence of choice and proof that this indeed defines a proper Dirac sequence.

We assume $0 \in \Omega$. Then there exists an $i \in \{0,\ldots,n\}$ with $0 \in [x_{i-1}, x_i)$. If not readily defined, we set $x_{i-2} = x_{i-1} - h \in \mathbb{R}$ and $x_{i+1} = x_i + h \in \mathbb{R}$. We define (see also Fig. 1)

$$\Phi_h(x) = \begin{cases} \frac{x_i}{h^3}x + \frac{x_i|x_{i-2}|}{h^3}, & x \in [x_{i-2}, x_{i-1}) \\ \frac{x_i+x_{i-1}}{h^3}x + x_i\frac{x_i+x_{i-1}}{h^3} - \frac{x_{i-1}}{h^2}, & x \in [x_{i-1}, x_i) \\ \frac{x_{i-1}}{h^3}x + \frac{x_{i+1}|x_{i-1}|}{h^3}, & x \in [x_i, x_{i+1}] \\ 0 & \text{else.} \end{cases} \text{, where } x, \Phi_h(x) \in \mathbb{R} \tag{36}$$

Then $\Phi_h$ is non-negative for all $h \in \mathbb{R}_{>0}$ and as a composition of integrable functions integrable with $\int \Phi_h(x)\,dx = 1$ (see App. B) and $\Phi_h \in L^1(\mathbb{R}) = \{f : \mathbb{R} \to \mathbb{R}; f \text{ integrable and } \int |f(x)|\,dx < \infty\}$.

Now, let $\rho \in \mathbb{R}_{>0}$ and set $h = \frac{\rho}{2} \in \mathbb{R}_{>0}$. Then we have by Eq. 36

$$\int_{\mathbb{R}\setminus\mathcal{B}_\rho(0)} \Phi_h\,dx = \int_{-\infty}^{\rho} \Phi_h\,dx + \int_{\rho}^{\infty} \Phi_h\,dx = \int_{-\infty}^{\rho} 0\,dx + \int_{\rho}^{\infty} 0\,dx = 0,$$

and it follows

$$\lim_{h\to0} \int_{\mathbb{R}\setminus\mathcal{B}_\rho(0)} \Phi_h\,dx = 0 \text{ for all } \rho \in \mathbb{R}_{>0}. \tag{37}$$

Hence Eq. 36 defines a Dirac sequence. We use $\Phi_h$ for the approximation of the $\delta$-distribution in our implementation of Equation 16.

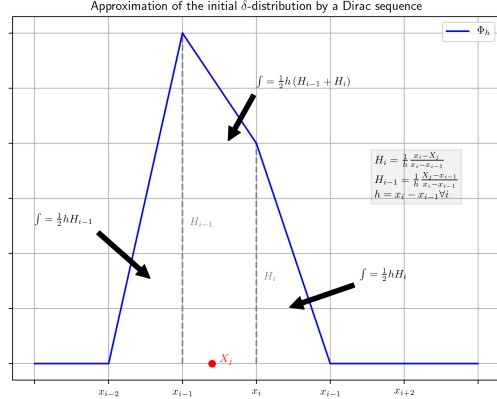

**Figure 1.** Dirac sequence $(\Phi_h)_{h \in \mathbb{R}_{>0}}$ for the approximation of the $\delta$-distribution in the initial value in Eq. 16. The function $\Phi_h$ depends on the spatial discretization fineness $h$ and converges to $\delta$ for $h \to 0$. The function $\Phi_h$ is piecewise linear with a peak at each data point $X_j, j \in \{1, ..., N\}$ integrating to 1.

The concept of the Dirac sequence also provides the justification to generally rely on the $\delta$-distribution in the construction of the initial value of the diffKDE. The Gaussian kernel defined in Eq. 10 that solves the diffusion equation as a fundamental solution is again a Dirac-sequence (Boccara, 1990). This link connects the diffKDE directly back to the $\delta$-distribution.

### 5.4 The diffKDE algorithm realization in Python

The implementation was conducted in Python and its concept shown in Alg. 1. We used the Python libraries Numpy (Harris et al., 2020) and SciPy (Virtanen et al., 2020; Gommers et al., 2022) and the Python Math module (Van Rossum, 2020) for data preprocessing, calculation of the bandwidths, setup of the differential equations and their solution. The algorithm iteratively calculates three KDEs: first the two for the approximations of $p$ and $f$ as the pilot estimation steps described in Sec. 4 and the last one being $u$ the solution of the diffKDE built on the two prior. All three KDEs are calculated by solving the diffusion equation up to the respective final iteration time. The solution is realized in *while*-loops solving Eq. 33. The two pilot estimation steps can be calculated simultaneoulsy, since they are independent of each other and only differ in their final iteration times $T_p$ and $T_f$. All input variables are displayed in Tab. 1 the return values listed in Tab. 2.

The spatial grid discretizing $\Omega$ is setup according to the description in Sec. 5.1 in lines 1 and 2 of Alg. 1. It consists of $n \in \mathbb{N}$ intervals, where $n$ can be set by the user. The boundary values are $x_{min} = \min X \in \mathbb{R}$ and $x_{max} = \max X \in \mathbb{R}$ by default, but can also be chosen individually. Setting the boundary values to an individually chosen interval in the function call results in a clipping of the used data to this smaller one before KDE calculation. Outside the interval boundaries, the diffKDE adds two additional discretization points to make it applicable for the case of a data point $X_j, j \in \{1, ..., N\}$ being directly located at one of the boundaries. This way it is possible to construct the initial value defined in Eq. 36, which takes into account the two

**Algorithm 1** Finite differences based algorithm for the implementation of the diffusion KDE.

Note: the routine solve($\mathbf{M}, \mathbf{b}$) means that the system $\mathbf{M}\mathbf{x} = \mathbf{b}$ is solved.

---

**Require:** $\boldsymbol{X} \in \mathbb{R}^N$, $n \in \mathbb{N}$, $timesteps \in \mathbb{N}$, $x_{min} \in \mathbb{R}$, $x_{max} \in \mathbb{R}$

1: $h \leftarrow (x_{max} - x_{min}) / (n - 4)$

2: $\boldsymbol{\Omega} \leftarrow (x_{min} - 2h, x_{min} - h, ..., x_{max} + h, x_{max} + 2h) \in \mathbb{R}^{n+1}$

3: $\boldsymbol{p}, \boldsymbol{f}, \boldsymbol{u} \leftarrow \Phi_h$

4: $T_p \leftarrow \sigma^2 \left( \frac{4}{3} N \right)^{-\frac{2}{5}}$

5: $T_f \leftarrow \left( 0.9 min \left( \sigma, \frac{iqr(data)}{1.34} \right) \right)^2 N^{-\frac{2}{5}}$

6: $t \leftarrow 0$, $\Delta_p \leftarrow T_p / timesteps$, $\Delta_f \leftarrow T_f / timesteps$

7: **while** $t < T_p$ **do**

8:     $\boldsymbol{p} \leftarrow \text{solve} \left( \mathbf{I}_{n+1} - \Delta_p \mathbf{A}_{pilot}, \boldsymbol{p} \right)$

9:     $\boldsymbol{f} \leftarrow \text{solve} \left( \mathbf{I}_{n+1} - \Delta_f \mathbf{A}_{pilot}, \boldsymbol{f} \right)$

10:    $t \leftarrow t + \Delta_p$

11: **end while**

12: $\boldsymbol{q} \leftarrow \sqrt{ \int_{\Omega} \left( \left( \frac{\boldsymbol{f}}{\boldsymbol{p}} \right)'' \right)^2 dh }$

13: $E_\sigma \leftarrow \frac{1}{n+1} \sum_{i=0}^{n+1} \sqrt{p(x_i)}$

14: $T \leftarrow \left( \frac{E_\sigma}{2N\sqrt{\pi}q^2} \right)^{\frac{2}{5}}$

15: $t \leftarrow 0$, $\Delta \leftarrow T / timesteps$

16: **while** $t < T$ **do**

17:    $\boldsymbol{u} \leftarrow \text{solve} \left( \mathbf{I}_{n+1} - \Delta \mathbf{A}, \boldsymbol{u} \right)$

18:    $t \leftarrow t + \Delta$

19: **end while**

20: **return** $\boldsymbol{u}, \boldsymbol{\Omega}, \boldsymbol{\Phi_h}, \boldsymbol{p}, \boldsymbol{stages}, \boldsymbol{times}$

---

neighbouring discretization points in each direction. This leads to a full set of $n + 1$ equidistant discretization points $(x_i)_{i=0}^n$ saved in a vector variable denoted in Alg. 1 as $\boldsymbol{\Omega}$. The spatial discretization $(x_i)_{i=0}^n$ includes an inner discretization between the handed in (or default set) interval endpoints $x_{min}$ and $x_{max}$ of $n - 4$ equally sized inner discretization intervals.

The Dirac sequence $\Phi_h$ for the implementation of the initial value is defined in Eq. 36 and we use the same for initialization of all three approximations of the the PDF ($p, f, u$) in line 3 of Alg. 1. In its calculation, the algorithm searches for each $j \in \{1, ..., N\}$ for the $i \in \{1, ..., n + 1\}$ with $x_i$ being the closest right neighbour of $X_j$. Then the initial value is constructed by assigning the values $\frac{1}{h} \frac{x_i - X_j}{x_i - x_{i-1}}$ and $\frac{1}{h} \frac{X_j - x_{i-1}}{x_i - x_{i-1}}$ at grid point $x_i$ and $x_{i-1}$, respectively, and zero elsewhere. These values are corresponding to the weighed heights $H_i$ and $H_{i-1}$ displayed in Fig. 1. The final initial value is the normalized sum of all these individual approximations of the $\delta$-distribution. All three used KDEs ($p, f, u$) are initialized with this initial value.

In the pilot estimation steps, we calculate the KDEs for $p$ and for $f$ required for the set up of the final iteration time $T^*$ for the diffKDE. The final iteration times $T_p$ and $T_f$ for $p$ and $f$, respectively, are calculated based on the input data $X$ in lines 4 and 5 of Alg. 1 as described in Sec. 4. Then the KDEs are calculated by solving a linear ordinary differential equation by an

**Table 1.** Input variables: The only required input variable for the calculation of the diffKDE is a one dimensional data set as an array like type. All other variables are optional, with some prescribed defaults. On demand the user can set individual lower and upper bounds for their data evaluation under the diffKDE as well as the number of used spatial and temporal discretization intervals. The individual selection of the final iteration time provides the opportunity to choose the specific smoothing grade on demand.

| index | name | type | default | description |
|---|---|---|---|---|
| 0 | $data$ | array like | required | input data $X \in \mathbb{R}$ |
| 1 | $xmin$ | float | $\min X$ | lower data boundary for KDE calculation |
| 2 | $xmax$ | float | $\max X$ | upper boundary for KDE calculation |
| 3 | $n$ | integer | 1004 | number of spatial discretization intervals in $\Omega$ |
| 4 | $timesteps$ | integer | 20 | number of temporal discretization intervals |
| 5 | $T$ | float | $T^*$ | final iteration time for diffKDE |

implicit Euler in the first *while*-loop in lines 7 to 9 of Alg. 1. For the pilot estimation steps calculating $p$ and $f$ the matrix $\mathbf{A}$ defined in Eq. 29 does not incorporate a parameter function and reduces to a matrix denoted as $\mathbf{A}_{pilot}$

$$\frac{1}{2}\frac{1}{h^2}\mathbf{V} = \mathbf{A}_{pilot} \in \mathbb{R}^{(n+1)\times(n+1)}, \tag{38}$$

where $\mathbf{A}_{pilot} \in \mathbb{R}^{(n+1)\times(n+1)}$ means that $A_{pilot}$ has real entries and $n+1$ rows and $n+1$ columns. Apart from this, the solutions for the pilot KDEs are the same as for the final diffKDE. The two pilot KDEs can be solved simultaneously, since they share their matrix $\mathbf{A}_{pilot}$ and have independent pre-computed final iteration times. The difference in their bandwidths is implemented in different time step sizes $\Delta_p$ and $\Delta_f$ for $p$ and $f$, respectively, which are initialized in line 6 of Alg. 1 directly before this
first *while*-loop. The temporal solutions are calculated $timesteps \in \mathbb{N}$ times in equidistantly increasing time steps until each individual final iteration time. Since we solve implicitly, there is no restriction to the time step size. But a larger $timesteps$ parameter reduces the numerical error proportional to the step size parameters $\Delta_p$ and $\Delta_f$. In this temporal solution we rely on the fact that the involved matrices are sparsely covered. The applied solver is part of the SciPy Python library and designed for efficient solution of linear systems including sparse matrices (Virtanen et al., 2020; Gommers et al., 2022).
The final iteration time $T$ for the diffKDE solution $u$ is calculated after the calculations of $p$ and $f$, using them both as described in Sec. 4 in lines 12 to 14 of Alg. 1. For the diffKDE $u$, the differential equation is given in Eq. 31 and the solution uses the implicit Euler approach in Eq. 33. This is implemented in a second *while*-loop described in lines 16 to 18 in Alg. 1, which is separated from the final iteration time $T^*$ and the matrix $\mathbf{A}$ identical to the calculations in the pilot step.
        The return value is a vector providing the user the diffKDE, along with some main parameters and the opportunity to also
evaluate different approximation stages. It provides in the first and second entry of the diffKDE and the spatial discretization $\Omega$. The third entry is the initial value $\Phi_\delta$ and the fourth pilot estimate $p$ that influences the adaptive smoothing. The last values to return are the two vectors $stages$ and $times$. These include the approximation stages of the diffKDE and the respective

**Table 2.** Return values of the diffKDE: The return variable of the diffKDE is a vector. Its first entry is the diffKDE evaluated on the spatial grid. Its second entry is the spatial grid $\Omega$.

| index | name | type | size | description |
|-------|------|------|------|-------------|
| 0 | $u$ | Numpy array | $n+1$ | diffKDE values on $\Omega$ |
| 1 | $\Omega$ | Numpy array | $n+1$ | spatial discretization |

times exceeding the default optimal solution stored in the diffKDE, providing some oversmoothed solutions for individual evaluations. The times are the 20 timesteps used for the calculation of $u$ as defined in Tab. 1 followed by additional 10 up to the doubled approximated optimal final iteration time $2T^*$. The time step size for the solutions between $T^*$ and $T^*$ are doubled because of the smaller changes in the solution for larger times as for example visible in Fig. 5.

Possible problems are caught in *assert* and *if* clauses. Initially, the data is reshaped to a Numpy array for the case of a list handed in and it is made sure that this is non-empty. For the case of numerical issues leading to a pilot estimate including zero values, the whole pilot is set back equal to 1 to ensure numerical convergence. Accordingly for the case of NaN value being delivered for the optimal bandwidth for the diffKDE, in which case this is also set to the bandwidth chosen for $f$ in Eq. 20.

### 5.5 Pre-implemented functions for visual outputs

Besides the standard use to calculate a diffKDE at an approximated optimal final iteration time for direct usage, we also included three possibilities to generate a direct visual output, one of them being interactive. Matplotlib (Hunter, 2007) provides the software measures for creating the plots. Most methods are part of the submodule Pyplot, whereas the interactive plot is based on the submodule Slider.

The function call *evol_plot* opens a plot showing the time evolution of the diffKDE (e.g., see Fig. 2 for example output). The plot includes drawings of the data points on the x-axis. In the background the initial values are drawn. The y-axis range is cut off at 20 % above the global maximum of the diffKDE to preserve focus of the graphic on the diffKDE and evolution. The evolution is presented by drawings of the individual time evolution stages using the sequential color map Viridis. The diffKDE is drawn in a bold blue line. This visualization of the evolution provides the user with insight into the data distribution and their respective influence on the final form of the diffKDE.

The function call *pilot_plot* opens that shows the diffKDE together with its pilot estimate $p$, showing the intensity of local smoothing (e.g., see Fig. 3 for example output). With this the user has the possibility to gain insight to the influence of this pilot estimator on the performance of the diffKDE. This plot also includes the data points on the x-axis.

The function call *custom_plot* opens an interactive plot, allowing the user to slide through different approximation stages of the diffKDE (e.g., see Fig. 4 for example output). This feature is based on the Slider module from the Matplotlib library (Hunter, 2007) and opens a plot showing the diffKDE. At the bottom of this plot is a scale that shows the time, initially being

set to the optimal iteration time derived from Eq. 17 in the middle of the scale. By clicking to the scale, the user can display the evolution stages at the respective (closest) iteration time. This reaches down to the initial value and up to the doubled optimal

iteration time. This interactive tool provides the user a simple tool to follow the estimate at different bandwidths. With the help of such plot it is possible to decide on whether the diffKDE is desired to be applied with a final iteration time that is different from the default.

## 6 Results on artificial data

In the following we document the performance of the diffKDE on artificial and real marine biogeochemical data. Whenever

not stated otherwise, we used the default values of the input variables stated in Tab. 2 in the calculation of the diffKDE.

For testing our implementation against a known true PDF, we first constructed a three-modal distribution. The objective is to assess the diffKDE's resolution and to exemplify the pre-implemented plot routines. The distribution was constructed from three Gaussian kernels centered around $\mu_1 = 3$, $\mu_2 = 6.5$ and $\mu_3 = 9$ and with variances $\sigma_1^2 = 1$, $\sigma_2^2 = 0.7^2$ and $\sigma_3^2 = 0.5^2$, each of them with a relative contribution of 30 %, 60 % and 10 %, respectively:

$$f(x) = 0.3 \frac{1}{\sqrt{2\pi}} e^{-\frac{1}{2}(x-3)^2} + 0.6 \frac{1}{0.7\sqrt{2\pi}} e^{-\frac{1}{2}\left(\frac{x-6.5}{0.7}\right)^2} + 0.1 \frac{1}{0.5\sqrt{2\pi}} e^{-\frac{1}{2}\left(\frac{x-9}{0.5}\right)^2}. \tag{39}$$

### 6.1 Pre-implemented outputs

As described in Sec. 5.5, we included three plot functions in the diffKDE implementation. All of them open pre-implemented plots, to give an impression of the special features that come with the diffKDE. An overview of the three possible direct visual outputs of the diffKDE software is described below.

First we demonstrate how to display the diffKDE's evolution. By calling the *evol_plot* function, a plot opens that shows all temporal evolution stages of the solution of Eq. 33. The temporal progress is visualized by a sequential colorscheme progressing from light yellow over different shades of green to dark blue. On the x-axis, all used data points are drawn and in the background a cut-off part of the initial value in light yellow as the beginning of the temporal evolution. The final diffKDE is plotted as a bold blue line in front of the evolution process. This gives the user an insight in the distribution of the initial data and

their influence on the shape of the estimate. As an example of the default setting, we created an evolution plot from 100 random samples of Eq. 39 visualized in Fig. 2. The second example shows the possibility of displaying the diffKDE together with the pilot estimate $p$ by the function *pilot_plot*. This is the parameter function in Eq. 16 responsible for the adaptive smoothing. Where this function is larger, the smoothing is less intense and allows more structure in the estimate of the diffKDE. Contrarily where it is smaller, the smoothing becomes more pronounced and data gaps are better smoothed out. The result of the diffKDE

is shown together with its parameter function $p$ in figure Fig. 3 on the same random sample of the distribution from Eq. 39 as before.

Lastly, we illustrate example snapshots of the interactive option to investigate different smoothing stages of the diffKDE by the function. We chose simpler and smaller example data for this demonstration, because these are better suited for visualization of this tool's possibilities. The function *custom_plot* opens an interactive graphic, starting with a plot of the approximated

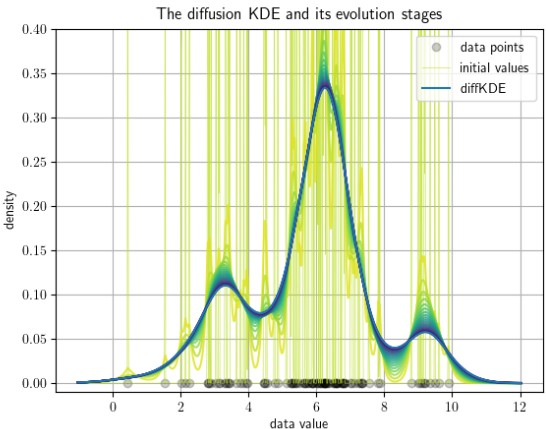

**Figure 2.** Pre-implemented direct visual output of the evolution process of the diffKDE. The input data are 100 samples randomly collected from Eq. 39. The individual data points are drawn on the x-axis. The y-axis represents the estimated probability density. The light yellow vertical lines in the background are the initial value of the the diffKDE. The temporal evolution of the solution of Eq. 33 is visualized by the sequent color scheme from light yellow over green to the bold blue graph in the front. The final diffKDE at the approximated optimal final iteration time represents as this graph the end of the time evolution.

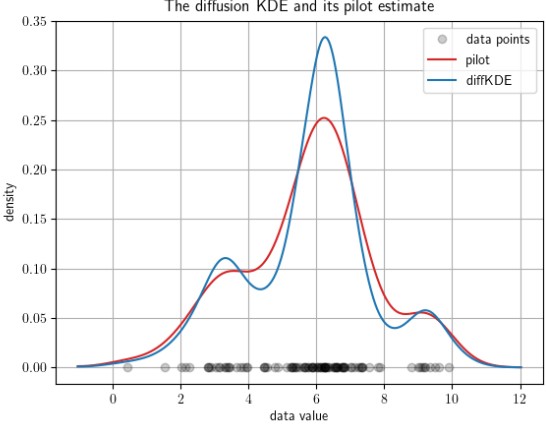

**Figure 3.** The diffKDE and its pilot estimate $p$. The input data are 100 samples randomly collected from Eq. 39. The data points are drawn on the x-axis. The y-axis represents the estimated density of the diffKDE in blue and the pilot estimate in red.

optimal default solution of the diffKDE at $T^*$. In this graphic the user is able to individually choose, by a slider, the iteration time at which the desired approximation stage of the diffKDE can be seen. The time can be chosen from 0, where the initial value is shown, up until the doubled approximated optimal time $(2 \times T^*)$. A reset button sets the graphic back to its initial stage of the diffKDE at $T^*$. Four snapshots of this interactive experience are drawn in Fig. 4.

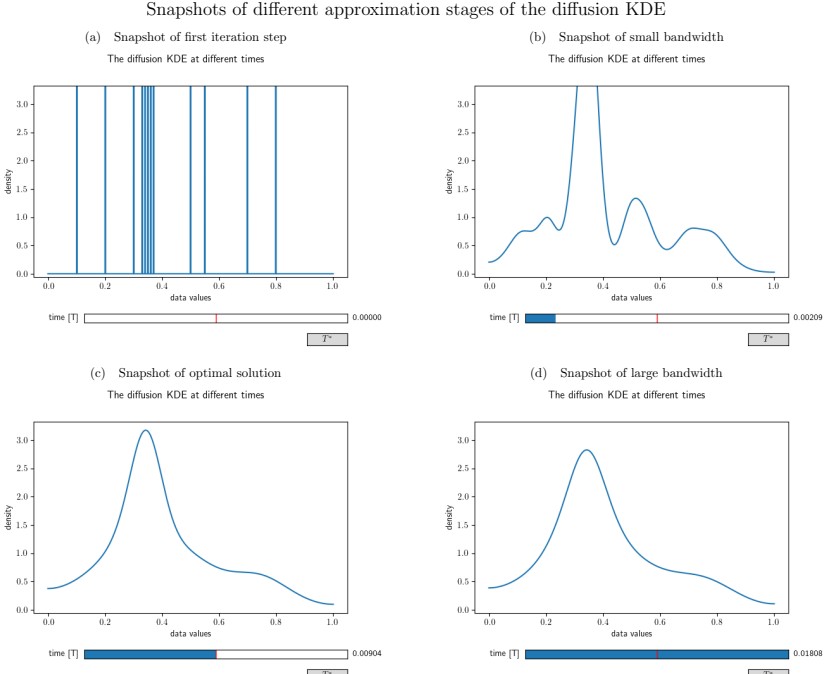

**Figure 4.** Different snapshots from the interactive visualization of the diffusion KDE generated from the artificial data set $(0.1, 0.2, 0.3, 0.33, 0.34, 0.35, 0.36, 0.37, 0.5, 0.55, 0.7, 0.8)$. (a) shows the output at $time = 0$ and hence the initial value. (b) shows an intermediate smoothing stage of the diffKDE. (c) shows the diffKDE of the input data at the approximated optimal iteration time $T^*$. This is the initial stage of the interactive graphic. By clicking the button on the lower right, the graphic can be reset to this stage. (d) shows an oversmoothed version of the diffKDE at the doubled approximated optimal iteration time.

## 6.2  Performance analyses on known distributions and in comparison to other KDEs

In this section we present results obtained by random samples of the trimodal distribution from Eq. 39 and lognormal distributions with differing parameters. Wherever suitable, the results are compared to other commonly used KDEs. These include the most common Gaussian KDE with the kernel function from Eq. 10 (Gommers et al., 2022), the Epanechnikov KDE with the kernel function from Eq. 9 (Pedregosa et al., 2012) and an improved implementation of the Gaussian KDE by Botev et al. (2010) in a Python implementation by Hennig (2021). We begin with an example of how the user may choose individually

different smoothing grades of the diffKDE, then compare the different KDEs with the true distribution, followed by investigating the influence of noise on different KDEs, and finally show the convergence of different KDEs to the true distribution with increasing sample size.

  We start with an individual selection of the approximation stages. This is one of the main benefits of the diffKDE compared to standard KDEs by providing naturally a family of approximations. This family can be observed by the function *custom_plot*.

Individual members can be produced by setting the bandwidth parameter $T$ in the function call of the diffKDE. This gives the

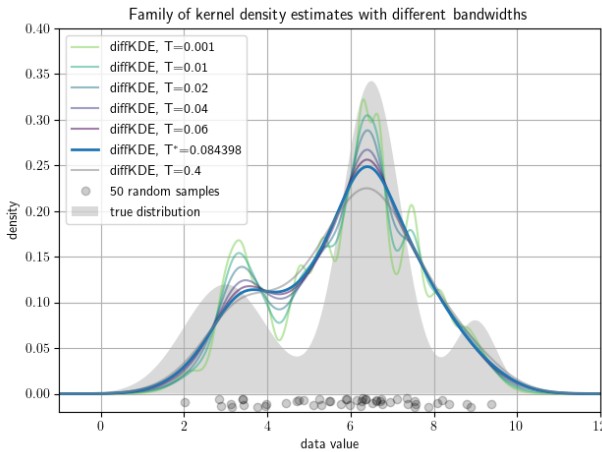

**Figure 5.** Family of diffKDEs evaluated at different bandwidths: A data set of 50 random samples drawn as grey circles on the x-axis serve to show the possibility to investigate a whole family of estimates by the diffDKE. The bold blue line represents the default solution of the diffKDE by solving the diffusion equation up to the approximated optimal final iteration time $T^*$. The other colors depict more detailed prior approximation stages with smaller bandwidth, i.e. earlier iteration times, and a smoother estimate with a far larger iteration time.

user the chance to choose among more and less smooth approximations. A selection of such approximations along with the default solution are shown in Fig. 5 on a random sample of 50 data points from the trimodal distribution in Eq. 39. The plot shows how smaller iteration times resolve more structure in the estimate, while a substantially larger iteration time has only little influence on the increased smoothing of the diffKDE.

In the following we only work with the default solution of the diffKDE at $T^*$. We start with comparisons of the diffKDE and the three other popular KDEs directly to the underlying true distribution. The three other KDEs are the Gaussian KDE in an implementation from SciPy (Gommers et al., 2022), the Epanechnikov KDE in an implementation from Scikit-learn (Pedregosa et al., 2012) and an improved Gaussian KDE by Botev et al. (2010) in a Python implementation by Hennig (2021).

We use differently sized random samples of the known distribution from Eq. 39 and the standard lognormal distribution both
over $[-1, 12]$, for a direct comparison of the accuracy of the KDEs. The random samples are 50, 100 and 1000 data points of each distribution and all four KDEs are calculated and plotted together in Fig. 6. The underlying true distribution is plotted in the background to visually assess the approximation accuracy. In general, the diffKDE resolves more of the details of the structure of the true distribution, while not being too sensitive to patterns introduced by the selection of the random sample and individual outliers. For the 50 random samples test of the trimodal distribution, all KDEs do not detect the third mode and only
the diffKDE and the Epanechnikov KDE detect the second. The magnitude of the main mode is also best resolved by these two. In the 100 random samples test of the trimodal distribution, the diffKDE and the Botev KDE are able to detect all three modes. The main mode is best resolved by the diffKDE, whereas the third mode best by the Botev KDE. In both test cases for the trimodal distribution, the Gaussian KDE is the smoothest and the Epanechnikov KDE provides the least smooth graph. In the 1000 random samples test the diffKDE best detects the left mode and the Botev KDE the two others best. Generally, diffKDE

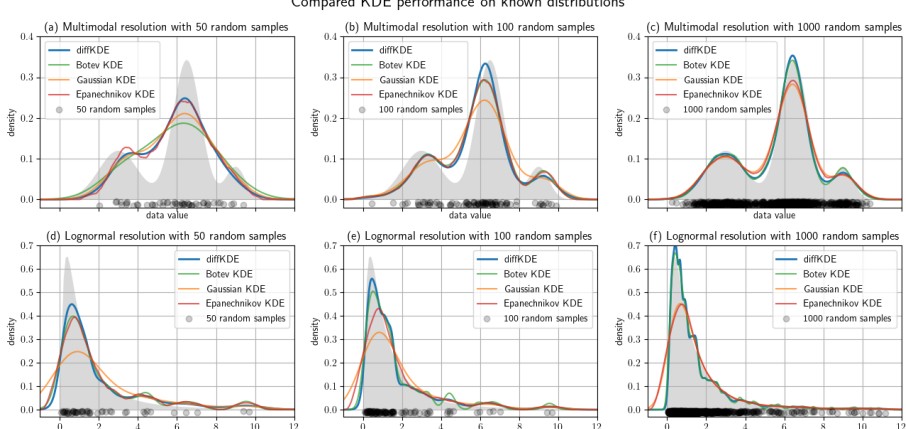

**Figure 6.** Test cases with known distributions: The plots (a), (b) and (c) show KDEs of random samples of the trimodal distribution defined in Eq. 39, (d), (e) and (f) the same for a lognormal distribution. The left figure column is constructed from 50 random samples, the middle from 100 and the right from 1000. In all plots the true distribution is drawn in grey in the background and the random data sample as grey dots on the x-axis. Each subfigure shows four KDEs: the diffKDE, the Botev KDE, the Gaussian KDE and the Epanechnikov KDE. In the labels of the KDEs are also the integrals over the interval $[-1, 12]$ given for each of the KDEs

and Botev KDE are closely aligned in this case. The Gaussian and Epanechnikov KDEs are also closely aligned, but with a worse fit of all structures of the true distribution. The steep decline to $0$ is best reproduced by the diffKDE particulary with low random sample sizes. The Gaussian KDE always performs the worst. The Botev KDE is generally also close to the diffKDE, but resolves in the tail of the distribution too much influence of individual outliers. In the 1000 random samples test with the lognormal distribution are again diffKDE and Botev KDE closely aligned as well as Gaussian and Epanechnikov KDE. The first two are very close to the true distribution, but resolve too much structure of the random sample. The diffKDE resolves more structure in the area close to $0$ and becomes smoother towards the tail of the distribution. The Botev KDE performs the other way around and provides a smoother estimate close to $0$ and more structure of the random sample towards higher data values. An analysis of the integral of the KDEs over the observed domain is presented in Tab. 3 and reveals that the diffKDE is the only one that integrates to $1$ in all test cases.

We refined the test cases from Fig. 6 by investigating a lognormal distribution with different parameters and a restriction to the interval $[0, 12]$ in Fig. 7. We varied mean and variances of the normal distribution and used two different means and three different variances resulting in six test cases. All of them are run with 300 random samples and again with all four KDEs. The larger the variance becomes, the more structure of individual data points is resolved by the Botev KDE. The Gaussian KDE fails for increasing variance too, resulting in intense oversmoothing. The Epanechnikov KDE performs well for smaller variances and larger means, but also oversmoothes in the other cases. The diffKDE is generally one of the closest to the true distribution, while not resolving too much of the structure introduced by the choice of the random sample, especially for

**Table 3.** Integrals of the KDEs displayed in Fig. 6 and Fig. 7.

| Graphic | diffKDE | Botev KDE | Gaussian KDE | Epanechnikov KDE |
|---|---|---|---|---|
| Figure 6 (a) | 1.0 | 0.999984 | 0.999984 | 0.999998 |
| Figure 6 (b) | 1.0 | 0.999999 | 0.999677 | 1.0 |
| Figure 6 (c) | 1.0 | 1.0 | 0.999989 | 1.0 |
| Figure 6 (d) | 1.0 | 0.999955 | 0.961448 | 0.999999 |
| Figure 6 (e) | 1.0 | 1.0 | 0.987128 | 0.999998 |
| Figure 6 (f) | 1.0 | 0.996 | 0.995571 | 0.996372 |
| Figure 7 (a) | 1.0 | 0.9986 | 0.894 | 0.9163 |
| Figure 7 (b) | 1.0 | 0.9094 | 0.802 | 0.8968 |
| Figure 7 (c) | 1.0 | 0.9999 | 0.9986 | 0.9617 |
| Figure 7 (d) | 1.0 | 1.0 | 0.9983 | 0.996 |
| Figure 7 (e) | 1.0 | 0.7703 | 0.0914 | 0.6171 |
| Figure 7 (f) | 1.0 | 0.6825 | 0.0337 | 0.5718 |

increased variances. But this too tends to resolve too much structure in the vicinity of the mode for smaller variances. The integrals of the KDEs are also presented in Tab. 3 and our implementation is again always exactly 1.

Now, we show the performance of the diffKDE on increasingly large data sets. We still use the trimodal distribution from Eq. 39. We start with four larger random data samples ranging from 100 to 10 million data points of the trimodal distribution and then being restricted to our core area of interest $[-1, 12]$. We calculate the diffKDE from all of them as well as the respective runtime on a consumer laptop from 2020. We compare the results again to the true distribution in Fig. 8. All of the estimates could be calculated in less than one minute. For 100 data points there is still an offset to the true distribution visible in the estimate. For the larger data samples the estimate only shows some minor uneven areas, which smooth out until the largest test case.

Furthermore, we investigated the convergence of the diffKDE to the true distribution, again in comparison to the three other KDEs. The error between the respective KDE and the true distribution is calculated by the Wasserstein distance (Panaretos and Zemel, 2019) with $p = 1$ by a SciPy function and the **MISE** defined in Eq. 4. For the approximation of the expected value in Eq. 4 we applied an averaging of the integral value for 100 different random samples for each observed sample size. We used increasingly large random samples from the trimodal distribution starting with 10 and reaching up to 1 million. The errors calculated for each of the KDEs on each of the random samples are listed in Tab. C1. The values from Tab. C1 are visualized in Fig. 9 on a log-scale. The diffKDE, the Gaussian and the Botev show a similar steep decline, while the Epanechnikov KDE far slower decreases its error with increased sample size. The diffKDE and the Botev KDE generally show similar error values,

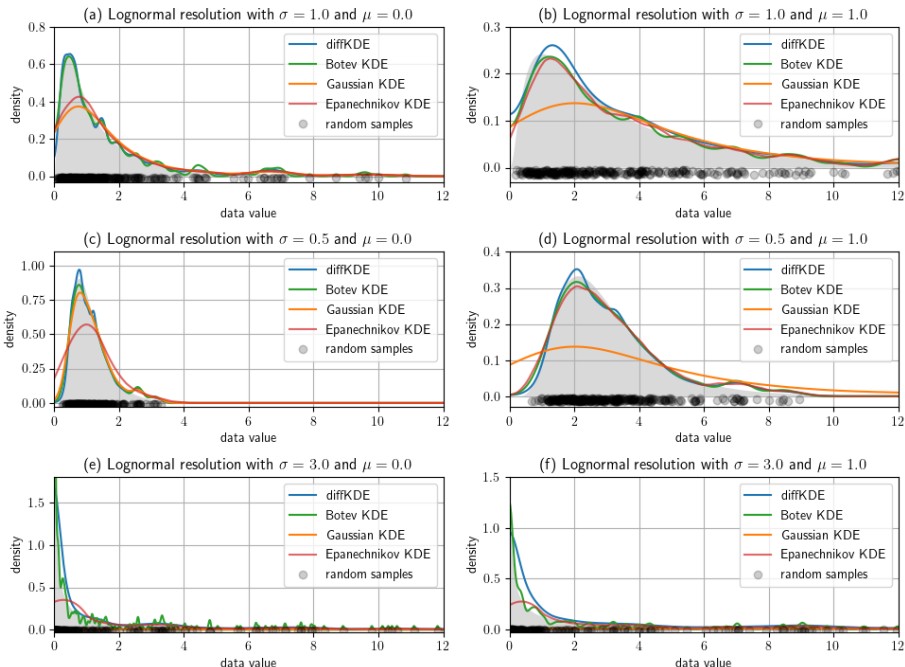

**Figure 7.** Lognormal test cases with different mean and variance parameters. Of each distribution 300 random samples were taken and the diffKDE, the Botev KDE, the Gaussian KDE and the Epanechnikov KDE calculated and plotted together with the true distribution. The random data sample is drawn as gray circles on the x-axis. (a) and (b) use $\sigma = 1$, (c) and (d) $\sigma = 0.5$ and (e) and (f) $\sigma = 3$ in their underlying normal distributions. The means are in the left column $\mu = 0$, the right column $\mu = 1$ of the underlying normal distribution.

the diffKDE relatively smaller ones on smaller data samples, the Botev KDE relatively smaller ones on data samples larger than around 5000.

Finally, we investigated the noise sensitivity of the diffKDE compared to the three other KDEs on data containing artificially introduced noise. We again used the trimodal distribution from Eq. 39 and 1000 random samples. From this, we created noised data $X_\theta \in \mathbb{R}^N$ by

$$(X_\theta)_i = (X)_i + (-1)^\tau \, rand \, 10^{-2} \theta \, \sigma \quad \text{for all } i \in \{1, ..., 1000\}, \tag{40}$$

where $\theta \in \{0, 1, 5, 15, 30\}$ defines the percentage of noise with respect to the standard deviation $\sigma \in \mathbb{R}$. $\tau \in \{1, 2\}$ was chosen randomly as well as $rand \in [0, 1]$. The error is again expressed by the Wasserstein distance between the original probability density and the respective KDE. The results are visualized in Fig. 10 with an individual panel for each KDE. The error of the Epanechnikov KDE is overall the largest and also increases to the largest. The Gaussian KDE produces the second largest error, but this even decreases with increased noise. The Botev KDE produces the smallest errors, but for increased noise this increases and approaches the magnitude of the one from the diffKDE. The error of the diffKDE only minimally responds to

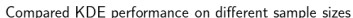

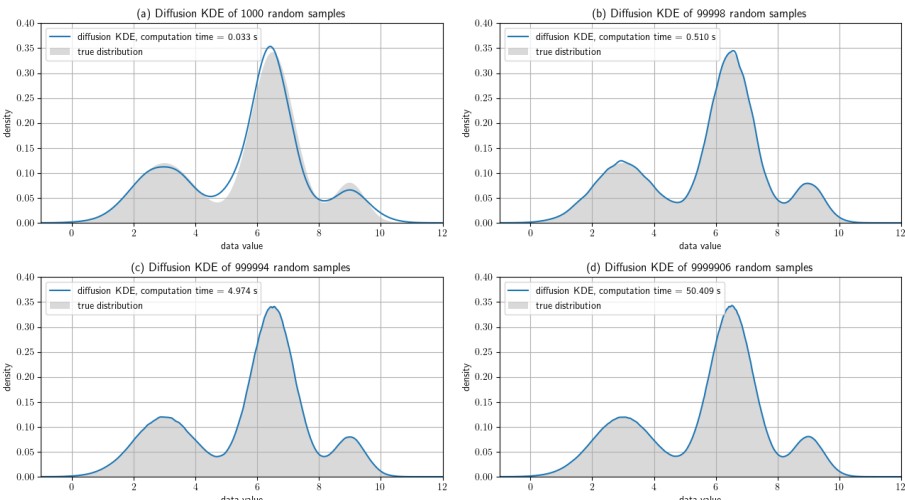

**Figure 8.** Test cases with different sample sizes: All four plots show the diffKDE of random samples of the known trimodal distribution defined in Eq. 39. (a) is calculated from a subsample of 100 data points, (b) $100,000$, (c) $1,000,000$ and (d) $10,000,000$, all cut to the interval $[-1, 12]$ and hence lacking a few data points. The true numbers of incorporated data points in the four test cases are given in the respective sub-headings. The measured computing time on a 2020 MacBook Air is also drawn in the respective label.

increased noise in the data. Visually, all four KDEs follow a similar pattern of a shift to the left of the graph. The Botev KDE additionally resolves more structure of the noised data as the noise increases.

## 7  Results on marine biogeochemical data and outlook to model calibration

The performance of the diffKDE is now illustrated with real data of a) measurements of carbon isotopes (Verwega et al., 2021;
Verwega et al., 2021), b) of plankton size (equivalent spherical diameter) (Lampe et al., 2021) and c) remote sensing data (Sathyendranath et al., 2019, 2021). We chose these data because we propose to apply the diffKDE for the analysis of field data for assessment and optimization of marine biogeochemical- as well as size-based ecosystem models. The carbon isotope data have been collected to constrain model parameter values of a marine biogeochemical model that incorporates this tracer as a prognostic variable (Schmittner and Somes, 2016).
Both data sets were already analyzed using KDEs in their original publications (Verwega et al., 2021; Lampe et al., 2021). Here we expand these analyses by a comparison of the KDEs used in the respective publications to the new implementation of the diffKDE. For the $\delta^{13}C_{POC}$ data, the Gaussian KDE was the one used in the data description publication. Since we have already done this in Sec. 6.2, we furthermore added the Epanechnikov and the Botev KDE to these graphics. For the plankton size spectra data, we only compared the diffKDE to the two Gaussian KDEs used in the respective publication to preserve the
clarity of the resulting figures.

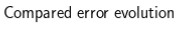

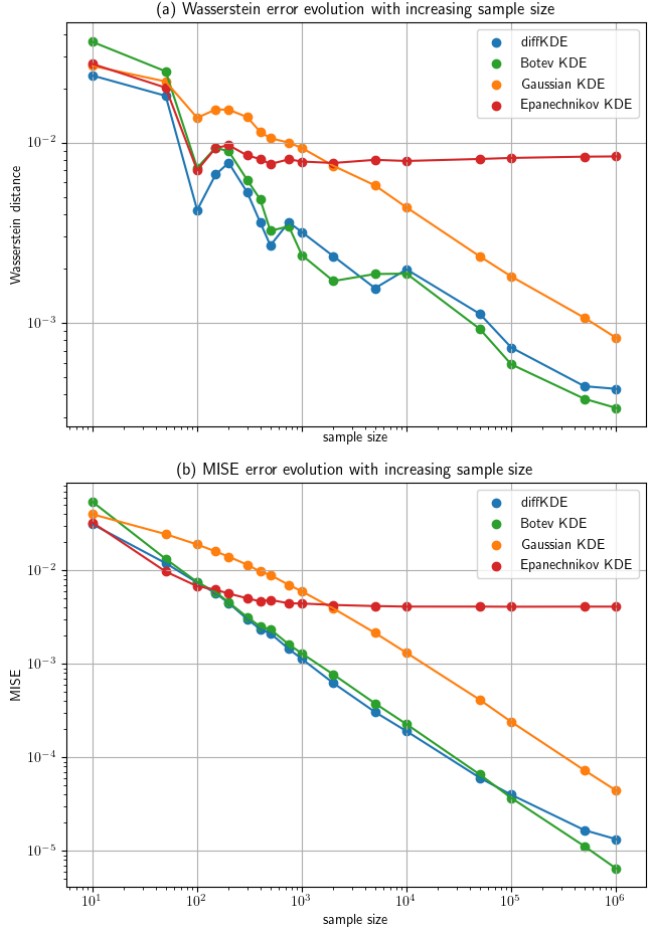

**Figure 9.** The evolution of the errors of the diffKDE, the Gaussian KDE, the Epanechnikov KDE and the Botev KDE are drawn on log-scale against the increasing sample size on the x-axis. (a) shows the error calculated with the Wasserstein distance and (b) with the MISE. The MISE is calculated after Eq. 4 from 100 different random samples.

## 7.1 Performance analyses on organic carbon-13 isotope data

The $\delta^{13}C_{POC}$ data (Verwega et al., 2021) was collected to serve for direct data analyses as well as for future model assessments (Verwega et al., 2021). We show here the Gaussian KDE as it was used in the data publication in a direct comparison to the diffKDE. Furthermore, we added the Epanechnikov and the Botev KDE. Since in this case no true known PDF is available, we have to compare the four estimates and subjectively judge their usefulness. In Fig. 11 we show the KDEs on four different subsets of the $\delta^{12}C_{POC}$ data: a) the full data set, b) a restriction to the core data interval of $[-35, -15]$, where $98.65$ % of the data is located, and then even further restricted to c) the euphotic zone and d) only data sampled in the 1990s. The euphotic zone describes the upper ocean layer with sufficient light to enable photosynthesis that produces organic matter

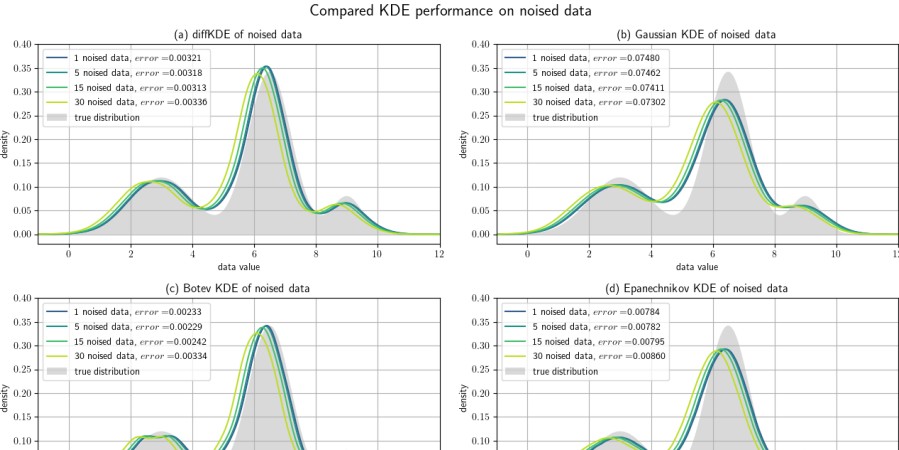

**Figure 10.** Noised data experiments: A random sample of 1000 data points of the trimodal distribution is artificially noised by differing amounts of the standard deviation. (a) shows the resulting diffKDEs of the differently noised data, (b) the Gaussian KDE, (c) the Botev KDE and (d) the Epanechnikov KDE. In all four panels the original true distribution is drawn in grey in the background. The values of the error between the KDEs and the original true distribution are also part of the respective labels.

(Kirk, 2011). While its depth can vary in nature (Urtizberea et al., 2013), here we pragmatically selected included data in
the upper 130 m consistent with the analysis in the data set description (Verwega et al. 2021). In all three cases that involve deep ocean measurements, the Botev KDE produces strong oscillations while the Gaussian KDE strongly smoothes the dip between the modes at around $\delta^{13}C_{POC} = -24$ and $\delta^{13}C_{POC} = -22$ and mostly the one between around $\delta^{13}C_{POC} = -28$ and $\delta^{13}C_{POC} = -24$. The Epanechnikov KDE resolves more structure than the Gaussian, but still less pronounced than the diffKDE. Especially in the full data analysis, the diffKDE reveals the most structure while not resolving smaller data features of individual
data points. The KDEs from the euphotic zone data are all reasonably smooth. The Gaussian KDE is again the smoothest and missing the mode at $\delta^{13}C_{POC} = -22$ completely. The other three KDEs resolve a similar amount of data structure. The Botev KDE reveals a better distinction between the modes at around $\delta^{13}C_{POC} = -24$ and $\delta^{12}C_{POC} = -22$ while the diffKDE shows the first one more pronounced. These observations are consistent with those from the experiments from Fig. 7 and Fig. Figure 10, where especially the Gaussian and the Botev KDE struggle with the resolution of data with increasing variances or
noise. From the four here observed $\delta^{12}C_{POC}$ data sets the euphotic zone data shown in panel (c) in Fig. 11 has with 7.78 the smallest standard deviation. The other shown data has variances 13.91, 10.96 and 9.61 for panels (a), (b) and (d), respectively.

### 7.2 Performance analyses on plankton size spectra data

Another example demonstrates the performance of the diffKDE if applied to plankton size data (Lampe et al., 2021). The data of size, abundance of protist plankton was originally collected for resolving changes in plankton community size-structure,

**Figure 11.** Comparison of KDE performance on marine biogeochemical field data: The $\delta^{13}C_{POC}$ data (Verwega et al., 2021) is in detail described in Verwega et al. (2021) and is covering all major world oceans, the 1960s to 2010s and reaches down into the deep ocean. In all four panels the diffKDE is plotted together with the Gaussian, the Epanechnikov and the Botev KDE. (a) Shows KDEs from all available data, (b) shows the KDEs of the data restricted to the core data values of $[-35, -15]$, (c) shows the KDEs from only euphotic zone data with values in $[-35, -15]$ and (d) the KDEs from all 1990s data with values in $[-35, -15]$.

providing complementary insight for investigations of plankton dynamics and organic matter flux (e.g., Nöthig et al., 2015). In the study of Lampe et al. (2021) a KDE was applied for the derivation of continuous size spectra of phytoplankton and microzooplankton that can potentially be used for the calibration and assessment of size-based plankton ecosystem models. In their study they used a Gaussian KDE, as proposed in Schartau et al. (2010), but with two different approaches for generating plankton size spectra. Uncertainties, also with respect to optimal bandwidth selection, were accounted for in both approaches by analyzing ensembles of pseudo-data resampled from original microscopic measurements. Smooth plankton spectra were obtained using the *combined* approach, where all phytoplankton and all zooplankton data were lumped together respectively and single bandwidths were calculated for every ensemble member (set of resampled data). This procedure avoided over-fitting but was also prone to over-smoothing, which can mask details, such as troughs in specific size ranges. More details in the size spectra were reolved with tan *composite* approach, where individual size spectra, calculated for each species or genus, were assembled. Since the variance within species or genus groups is smaller than within the large groups 'phytoplankton' or 'zooplankton', the individual bandwidths, and therefore the degree of smoothing, were considerably smaller than obtained in the combined approach. This computationally expensive method revealed many details in the spectra, but at the same time tended to resolve narrow peaks that were either clearly insignificant or remained difficult to interpret (see supplemental material in Lampe et al. (2021)). The here proposed diffKDE is tested with resampled data used for the simpler *combined* approach. The objective is to identify details in the size spectra that remained previously unresolved while insignificant peaks,

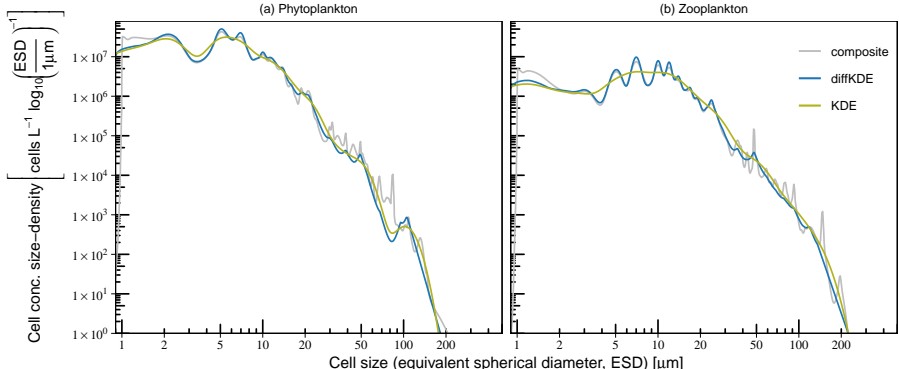

**Figure 12.** Comparison of KDE performance on (a) phytoplankton and (b) microzooplankton size spectra. The construction of composite and combined size spectra is described in Lampe et al. (2021) and based on Gaussian KDEs.Smoother combined spectra are the result of one KDE with a common bandwidth for all data. More structured composite spectra were assembled from taxon-specific spectra with individual, hence smaller, bandwidths.

as found in the composite approach, become smoothed out. Figure 12 shows the performance of the diffKDE in comparison to the original combined and composite spectra that were derived as ensemble means of estimates obtained with a Gaussian KDE. The spatial discretization of the diffKDE was set to $n = 1000$ to be comparable to the other already published KDEs in this case. The diffKDE seems to meaningfully combine the advantages of the two Gaussian KDE approaches in both
spectra, of the phytoplankton and microzooplankton respectively. With the diffKDE it is possible to generate estimates that display more detailed structure of the composite KDE. In some size ranges, such as between 5 and 50 $\mu$m, in particular in the microzooplankton spectrum. Concurrently, detailed variations, as caused by overfitting in the composite spectra, become suppressed for cell sizes larger than 10 $\mu$m. Thus, with the diffKDE it is possible to generate a single robust estimate that otherwise is only achieved by analyzing a composite of a series of individual estimates of a Gaussian KDE. The application of
the diffKDE for analysing details in plankton size spectra, or generally in particle size spectra, reduces computational efforts considerably.

### 7.3 Performance analyses on remote sensing data

Our last example refers to PDFs that reflect temporal changes (monthly means) of surface chlorophyll-a concentration within an off-shore ocean region (approximately 350 km × 330 km) that exhibits substantial mesoscale and sub-mesoscale variability.
The selected area is part of the of the Mauritanian upwelling system, located at the Moroccan coast of North Africa. This eastern boundary upwelling is known for the formation and spread of filaments, with some distinct characteristics in terms of the spatial variability in temperature and the distribution of plankton (e.g., Sylla et al., 2019; Romero et al., 2020; Versteegh et al., 2022). For this example we use remote sensing (satellite) data of monthly mean chlorophyll-a concentration from the year 2019, as processed and made available through the Ocean-Colour Climate Change Initiative (OC-CCI) (Sathyendranath

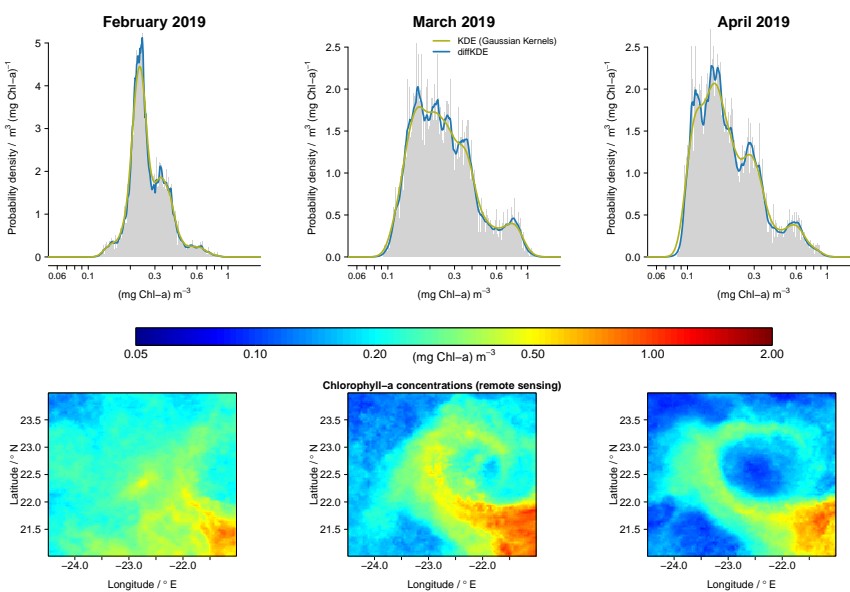

**Figure 13.** Comparison of KDE performance using monthly means (February, March, and April) of chlorophyll-a concentration derived from remote sensing data of the year 2019. The PDFs (upper panel) represent the temporal development of spatial variability seen in a subregion of the Mauritanian upwelling system (lower panel).

et al., 2019, 2021). We deliberately chose three months in which an eddy-like filamentous structure of elevated chlorophyll-a concentration developed after February, and then evolved during a two months period (March and April). Such development reveals specific spatial patterns, of low and increased chlorophyll-a, which leave a clear imprint in the corresponding PDFs, as depicted in Figure 13. In Figure 13 we find multiple modes, as well as a shift towards low chlorophyll-a concentrations that are much better resolved when the diffKDE is applied. Such details derived with the diffKDE could be compared with complementary PDFs, for example as obtained from remote sensing sea surface temperature data within the same time period, and we might gain further insight into the underlying processes involved in generating distinctive spatial and temporal patterns. Also, in case of simulations of mesoscale and sub-mesoscale processes, we should not expect to obtain a model solution with filamentous structures at identical times at the same places. But, we may regard a model's performance as credible if the model's solution yields similar spatial structures visually within the same region, perhaps at some different time, and the associated PDFs may then be directly assessed against the PDFs obtained from the remote sensing data.

### 7.4 Future application to model assessment and calibration

In geoscientific research, the derivation and comparison of well resolved PDFs can be useful, as demonstrated in our selected examples. Yet, the significance of resolving details in nonparamteric PDFs remains unclear. However, having high resolution PDFs available, as obtained with the diffKDE, is readily of value, and will likely guide further research.An obvious benefit of the diffKDE is its lesser dependence on the specification of a single, albeit optimal, bandwidth. Its application is likely more

robust for the assessment of simulation results, either against data or results of other models (e.g. multi model ensembles), which is particularly relevant for evaluations of future climate projections obtained with Earth system models (e.g., Oliver et al., 2022). The presented diffKDE provides a nonparametric approach to estimate PDFs with typical features of geoscientific data. Being able to resolve typical patterns such as multiple or boundary close modes, while being insensitive to noise and

individual outliers makes the diffKDE a suitable tool for future work in the calibration and optimization of Earth system models.

Comparison of model and field data requires additional processing to account for spatial-temporal differences between collected samples and model resolution. Typically, simulation results are available at every single spatial grid point and in every time step. In comparison, field data are usually sparsely available only. Interpolating such sparse field data can introduce

high uncertainty (e.g., Oliver et al., 2022). PDFs provide a useful approach to investigate data independent of the number of data points available (Thorarinsdottir et al., 2013). A comparison of two such functions can easily resolve the issue of non-equal field observations and simulation results. Histograms are commonly used as an approach to compare and ultimately constrain the distribution of model data to observations. However, many issues arrive including the subjective selection of intervals and histograms not being proper PDFs themselves.

The presented diffKDE provides a nonparametric approach to estimate PDFs with typical features of geoscientific data. Being able to resolve typical patterns such as multiple or boundary close modes, while being insensitive to noise and individual outliers makes the diffKDE a suitable tool for future work in the calibration and optimization of Earth system models.

## 8   Summary and conclusions

In this study we constructed and tested an estimator (KDE) of probability density functions (PDFs) that can be applied for

analysing geoscientifc and ecological data. KDEs allow the investigation of data with respect to their probability distribution, and PDFs can be derived even for sparse data. To be well suited for geoscientific data, the KDE must work fast and reliably on differently sized data sets, while revealing multimodal details as well as features nearby data boundaries. A KDE should not be overly sensitive to noise introduced by measurement errors or by numerical uncertainties. Such an estimator can be applied for direct data analyses or can be used to construct a target function for model assessment and calibration.

We presented a novel implementation of a KDE based on the diffusion heat process (diffKDE). This idea was originally proposed by Chaudhuri and Marron (2000) and its benefits in comparison to traditional KDE approaches were widely investigated by Botev et al. (2010). We chose this approach to KDE, because it offers three main benefits: (1) consistency at the boundaries (2) better resolution of multimodal data (3) a family of KDEs with different smoothing intensities. We provide our algorithm in an open source Python package. Our approach includes a new approximation of the bandwidth, which equals the

square root of the final iteration time. We directly approximate the analytical solution of the optimal bandwidth with two pilot estimation steps and finite differences. We calculate the pilot estimates as solutions of a simplified diffusion equation up until final iteration times derived from literature based bandwidths called *rule of thumb* by Silverman (1986). Our new approach results in three subsequent estimations of the PDF, each of them chosen with a finer bandwidth approximation.

Finite differences build the fundamentals of our discretization. The spatial discretization are equidistant finite differences. The $\delta$-distribution in the initial value is discretized by piecewise linear functions along the spatial discretization points constructing a Dirac-sequence. For the timestepping we applied an implicit Eulerian algorithm on an ordinary differential equation set up by a tridiagonal matrix corresponding to the diffusion equation on the spatial equidistant grid.

Our diffKDE implementation includes pre-implemented default output options. The first is the visualization of the diffusion time evolution showing the sequence of all solution steps from the initial values to the final diffKDE. This lets a user view the influence of individual data points and outlier accumulations on the final diffKDE and how this decreases over time. The second is the visualization of the pilot estimate that is also included in the partial differential equation to introduce adaptive smoothing properties. This provides the user an easy insight into the adaptive smoothing as well as the lower boundary of structure resolution given by this parameter function. Finally, an interactive plot provides a simple opportunity to explore all of these time iterations and look even beyond the optimal bandwidth and see smoother estimates.

Our implementation is fast and reliable on differently sized and multimodal data sets. We tested the implementation for up to 10 million data points and obtained acceptably fast results. A comparison of the diffKDE on known distributions together with classically employed KDEs showed reliable and often superior performance. For comparison we chose a SciPy implementation (Gommers et al., 2022) of the most classical Gaussian KDE (Sheather, 2004), an Scikit implementation (Pedregosa et al., 2012) of an Epanechnikov KDE (Scott, 1992) and a Python implementation (Hennig, 2021) of the improved Gaussian KDE developed by Botev et al. (2010). We designed multimodal and different boundary-close distributions and found our implementation to generate the most reliable estimates across a large range of sample sizes (Fig. 9). The diffKDE was neither prone to oversmoothing nor overfitting of the data, which we could observe in the other tested KDEs. A noise sensitivity test in comparison to the other KDEs also showed a good stability of the diffKDE against noise in the data.

An assessment of the diffKDE on real marine biogeochemical field data in comparison to usually employed KDEs reveals superior performance of the diffKDE. We used carbon isotope and plankton size spectra data and compared the diffKDE to the KDEs that were used to explore the data in the respective original data publications. On the carbon isotope data, we furthermore applied all previous KDEs for comparison. In both cases we were able to show that the diffKDE resolves relevant features of the data while not being sensitive to individual outliers or uncertainties (noise) in the data. We were able to obtain a best possible and reliable represantation of the true data distribution, better than those derived with other KDEs.

In future studies the diffKDE may potentially be used for the assessment, calibration and optimization of marine biogeochemical- and Earth system models. Already a plot of PDFs, of field data and simulation results respetcively, may provide visual insight into some shortcomings of the applied model. A target function can be constructed by adding a distance like the Wasserstein distance (Panaretos and Zemel, 2019) or other useful metrics for the calibration of climate models that can be investigated (Thorarinsdottir et al., 2013). Thus, KDE applications such as our diffKDE can greatly simplify comparisons of differently sized field and simulation data sets.

*Code availability.* The exact version of the diffKDE implementation (Pelz and Slawig, 2023) used to produce the results used in this paper is archived on Zenodo: https://doi.org/10.5281/zenodo.7594915.

## Appendix A: Optimal bandwidth choice for the general KDE

The derivation of the optimal bandwidth choice for a KDE was already described in Parzen (1962) and can be found in more detail in Silverman (1986). The additional conditions stated in Eq. 7 to the kernel function

$$\int_{\mathbb{R}} y K(y)\, dy = 0 \text{ and } \int_{\mathbb{R}} y^2 K(y)\, dy = k_2 \in \mathbb{R} \setminus \{0\} \tag{A1}$$

correspond to the order of the kernel being equal to 2 (Berlinet, 1993). For such kernels Silverman (1986) showed the minimizer of the asymptotic mean integrated squared error to be

$$h = \left( \frac{\int_{-\infty}^{\infty} K(y)^2\, dy}{N k_2^2 \int_{-\infty}^{\infty} f''(x)^2\, dx} \right)^{\frac{1}{5}} = \left( \frac{\|K\|_{L^2}^2}{N k_2^2 \|f''\|_{L^2}^2} \right)^{\frac{1}{5}} \tag{A2}$$

In our context of working with the squared bandwidth $t = h^2$ this optimal bandwidth choice becomes

$$t = \left( \frac{\|K\|_{L^2}^2}{N k_2^2 \|f''\|_{L^2}^2} \right)^{\frac{2}{5}}, \tag{A3}$$

which equals Eq. 13.

## Appendix B: Integral property of the Dirac sequence

Here, we briefly give the proof of the integral property of the used Dirac sequence $\Phi_h$ defined in Equation 36. Let $h \in \mathbb{R}_{>0}$. Then we obtain

$$
\begin{aligned}
\int \Phi_h(x)\, dx &= \int_{x_{i-2}}^{x_{i-1}} \Phi_h(x)\, dx + \int_{x_{i-1}}^{x_i} \Phi_h(x)\, dx + \int_{x_i}^{x_{i+1}} \Phi_h(x)\, dx \\
&= \frac{1}{2}(x_{i-2} - x_{i-1}) \frac{1}{x_{i-2} - x_{i-1}} \frac{x_i}{x_i - x_{i-1}} + \frac{1}{2}(x_i - x_{i-1})\left( \frac{1}{x_{i-2} - x_{i-1}} \frac{x_i}{x_i - x_{i-1}} + \frac{1}{x_{i+1} - x_i} \frac{-x_{i-1}}{x_i - x_{i-1}} \right) \\
&\quad + \frac{1}{2}(x_{i+1} - x_i) \frac{1}{x_{i+1} - x_i} \frac{-x_{i-1}}{x_i - x_{i-1}} \\
&= \frac{1}{2}h\frac{1}{h}\frac{x_i}{h} + \frac{1}{2}h\left( \frac{1}{h}\frac{x_i}{h} + \frac{1}{h}\frac{-x_{i-1}}{h} \right) + \frac{1}{2}h\frac{1}{h}\frac{-x_{i-1}}{h} \\
&= \frac{1}{2}\frac{x_i}{h} + \frac{1}{2}\frac{x_i}{h} - \frac{1}{2}\frac{x_{i-1}}{h} - \frac{1}{2}\frac{x_{i-1}}{h} \\
&= \frac{x_i - x_{i-1}}{h} \\
&= 1.
\end{aligned}
\tag{B1}
$$

**Table C1.** Error convergence of the observed KDEs in Fig. 9: The first column lists the sample sizes used for the calculation in each row. The following four lines show the error between the four observed KDEs and the true distribution calculated by the Wasserstein distance. The final four columns contain the equivalent errors, but calculated by the MISE.

| sample size | $W_{diffKDE}$ | $W_{BKDE}$ | $W_{GKDE}$ | $W_{EKDE}$ | $MISE_{diffKDE}$ | $MISE_{BKDE}$ | $MISE_{GKDE}$ | $MISE_{EKDE}$ |
|---|---|---|---|---|---|---|---|---|
| 10 | 0.0235 | 0.0362 | 0.0266 | 0.0273 | 0.0313 | 0.0544 | 0.04 | 0.0327 |
| 50 | 0.0181 | 0.0248 | 0.0218 | 0.0202 | 0.012 | 0.0133 | 0.0246 | 0.0098 |
| 100 | 0.0042 | 0.0072 | 0.0137 | 0.007 | 0.0074 | 0.0075 | 0.019 | 0.0068 |
| 150 | 0.0066 | 0.0094 | 0.0153 | 0.0093 | 0.0057 | 0.0059 | 0.016 | 0.0062 |
| 200 | 0.0078 | 0.0089 | 0.0152 | 0.0097 | 0.0045 | 0.0046 | 0.0141 | 0.0057 |
| 300 | 0.0053 | 0.0062 | 0.0139 | 0.0085 | 0.003 | 0.0032 | 0.0116 | 0.0051 |
| 400 | 0.0036 | 0.0048 | 0.0115 | 0.0081 | 0.0024 | 0.0025 | 0.0098 | 0.0047 |
| 500 | 0.0027 | 0.0032 | 0.0106 | 0.0076 | 0.0021 | 0.0024 | 0.009 | 0.0049 |
| 750 | 0.0036 | 0.0034 | 0.01 | 0.0081 | 0.0015 | 0.0016 | 0.007 | 0.0045 |
| 1000 | 0.0032 | 0.0024 | 0.0093 | 0.0079 | 0.0011 | 0.0013 | 0.006 | 0.0044 |
| 2000 | 0.0024 | 0.0017 | 0.0074 | 0.0077 | 0.0006 | 0.0008 | 0.0039 | 0.0043 |
| 5000 | 0.0015 | 0.0019 | 0.0058 | 0.008 | 0.0003 | 0.0004 | 0.0022 | 0.0042 |
| 10000 | 0.002 | 0.0019 | 0.0044 | 0.0079 | 0.0002 | 0.0002 | 0.0013 | 0.0041 |
| 50000 | 0.0011 | 0.0009 | 0.0024 | 0.0081 | 0.00006 | 0.00007 | 0.0004 | 0.0041 |
| 100000 | 0.0007 | 0.0006 | 0.0018 | 0.0082 | 0.00004 | 0.00004 | 0.0002 | 0.0041 |
| 500000 | 0.0005 | 0.0004 | 0.0011 | 0.0084 | 0.00002 | 0.00001 | 0.00007 | 0.0041 |
| 1000000 | 0.0005 | 0.0003 | 0.00081 | 0.0084 | 0.00001 | 0.000007 | 0.00004 | 0.0041 |

## Appendix C:  Error convergence of observed KDEs

Tab. C1 shows the error values calculated by the Wasserstein distance and the MISE between the true distribution and the respective KDEs. The used distribution is the trimodal from Eq. 39 and the values plotted in Fig. 9.

*Author contributions.*  MTP set up the manuscript and developed the implementation of the diffusion-based kernel density estimator. VL conducted the comparison experiments with the plankton size spectra. MS conducted the comparison experiments with the remote sensing data. CJS edited the manuscript. MS and TS edited the manuscript and supported the development of the implementation of the diffusion-based kernel density estimator.

*Competing interests.* The contact author has declared that neither they nor their co-authors have any competing interests.

*Acknowledgements.* The editor and the authors thank two anonymous reviewers for their comments on this article. The first author is funded through the Helmholtz School for Marine Data Science (MarDATA), Grant No. HIDSS-0005.

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
