# Peer review of "A diffusion-based kernel density estimator (diffKDE, version 1) with optimal bandwidth approximation for the analysis of data in geoscience and ecological research"

_Geoscientific Model Development, 2023_

## Referee Comment (RC1)

**General Comments**

The authors present a novel kernel density estimation approach that combines a diffusion-based method with a pilot step to produce better estimates of the bandwidth - a classically difficult problem. The authors also develop a Python package, *diffKDE*, for easy use of their method and highlight some of the package's capabilities within the manuscript. diffKDE is benchmarked against pre-existing KDE methods using synthetic data and applied to real marine data, highlighting its applicability. However, the material is organized and presented in such a manner that detracts from its scientific contribution. Additionally, as noted in the points below, the verbiage, lack of technical definitions in multiple locations throughout the text, and sectioning of the ideas muddle the presentation of the material. I recommend the authors revise the manuscript and give special attention to the structure of the manuscript and the presentation clarity. I believe this work has potential and I look forward to the reading the revised manuscript.

**Specific Comments**

1. The organization of the paper detracts from the scientific contribution. Here are a couple examples:

   (a) Section 2.1 gives background on the general kernel density estimator *and* the proposed diffusion-based kernel density estimator with pilot study. Then section 2.2 and 2.3 discuss discretization of the diff-KDE and implementation, respectively. This feels out of order. I would assume general background would be its own section (e.g., make section 2.1 its own section) and the proposed method and *all* associated ideas would be grouped together (e.g., make sections 2.1.2 through 2.4 its own section, say section 3, and order these appropriately).

   (b) Regarding the order material is presented, there is no initial explanation as to why a pilot estimation is needed. It feels as though section 2.1.4 needs some prior context or motivation.

   (c) Section 3 contains the simulation example and real world example. I would makes these distinct sections (e.g., following the above point, sections 4 and 5). To this point, the first paragraph of section 3 is confusion. There is no clear distinction between the simulation explanation and the real world example. By separating these, that should mitigate any confusion.

   I strongly recommend reorganizing the paper by grouping similar ideas by sections with appropriate sub-sections.

2. Some of the verbiage and notation used in this paper may preclude the general audience of GMD from understanding its scientific merit. There are some examples included in the list below, but I recommend the authors check the manuscript and make sure *all* mathematical terms are defined and avoid using overly technical notation if possible.

3. Abstract: Some of the sentences need some work. For example:

(a) Line 1: A PDF is a function defining the probability of a random variable taking on a specific value. I don't like the use of observed or simulated variables or the phrase, "comprise basic information", as I think this is a little too weak of wording.

(b) Line 6: Starting with "A diffusion-based KDE..." Why is this the case? Maybe rephrase to say, "Diffusion-based KDEs have been shown to provide a useful approach ..."

(c) Line 10: What is boundary close data and what details are suppressed? Do you mean your approach produces a smooth surface that is robust to noise and outliers? Also, there are various other places in the manuscript that use "details", but it is unclear what is being referred to.

4. The introduction falls a short. There is a nice body of literature on diffusion-based KDE and it would be beneficial to this manuscript if it is put into context with the current literature.

5. Line 19: Any citations to support the claim for the need in other model applications? This seems like a great place to include more references.

6. Lines 24-27: Your definition of a PDF does not seem correct. You need to explicitly define each element of you probability space $\Omega, \mathcal{A}, P$. Also, I assume $\mathcal{A}$ is a $\sigma$-algebra and $\Omega$ is the sample space. In which case I think you mean to have $f : \Omega \to [0, \infty)$ (also, open bracket on the right). Also, the sentence, "By definition, ..." should be reworded.

7. Line 41: I like the list, but are there any more current (last 5 or so years) papers that could be included? A cursory literature search returned quite a few recent papers on the topic.

8. Line 48: Define $\delta$-distribution, or say something like, "which will be formally defined in section xxx.".

9. Line 75: Are all $X_j$ assumed to come from the same distribution or are they completely independent? Should they be independent or independent and identically distributed?

10. Line 78: Is $n$ the same as $N$? If so, verify the use of $n$ and $N$ throughout the paper.

11. Line 78 and generally: This is a personal choice, but I feel your paper would be more accessible to the general GMD audience if you change your function notation. Consider

$$\hat{f}_n(x; h) = \frac{1}{nh} \sum_j^n K\left(\frac{x - X_j}{h}\right) \quad \text{where } \hat{f} : \mathbb{R} \times \mathbb{R}_{>0} \to \mathbb{R}_{\geq 0}.$$

12. Line 86: Should it be $\mathbb{R}_{>0}$ or $\mathbb{R}_{\geq 0}$?

13. Line 91: It is not clear where equation (6) comes from. It would be helpful if this result, or derivation, is included in the appendix.

14. Lines 98 and 100: see comment 11.

15. Line 109: Is the diffusion based approach an alternative method? Better? Also, different approach in what sense? Make this more clear.

16. Line 110: What is "It"? The KDE? Also, "progresses up to an estimate at the final time," is unclear. What progresses and what estimate?

17. Line 113: see comment 11.

18. Line 121: What is the importance of acting inversely proportional to the diffusion quotient? Expand on this point.

19. Line 136: How do you set $p$ properly? It is not clear at this point. (E.g., Line 166 starting with "Choosing $p$ to be ..." could be incorporated here)

20. Line 147: Similar to comment 13, it is not clear where this equation comes from, consider adding its derivation to the appendix. Also, define all the terms in the equation (e.g., ... where $\| \cdot \|_{L^2}$ is the $L^2$-norm...).

21. Line 153: How or why is the additional effort avoided in your approach?

22. Line 168: What is "resolves the unexpected structure"

23. Line 185: The second equal should be an approximate sign. Same with multiple places following. Eq 15, 16, ...

24. Line 201: Bold $u$?

25. Line 208: Why not an explicit? Does the solver have stability issues? Is this relevant?

26. Line 220: Define $\mathcal{B}_\rho(0)$.

27. Line 222: How is $\frac{|\Omega|}{n} = h$ calculated? Moreover, isn't $\Omega = \mathbb{R}$?

28. Line 228: Is $L^1(\mathbb{R})$ defined?

29. Line 259: Define $iqr$. A non-statistical audience may not know what this is.

30. Line 271: Is the discretization of $\Omega$ the same as $\bar{\Omega}$? See line 180. Also, if this is the case, there are multiple instances throughout the paper where this appears.

31. Line 292: What do you mean by the variable $\Omega$? Is $\Omega$ the spatial domain or something else?

32. Table 1: Why is the default number of spatial discretization intervals 1004 and not some value dependent on the dimension of $\Omega$ or the concentration and number of observed points?

33. Line 309: What is time forward?

34. Line 324: Why 20 and 10? Any justification?

35. Section 2.4: I think this section could use a major rework to make it more readable. Also, there are figures that relate to each of these functions. Why not relate the function to its corresponding figure? For example, "The function call *evol_plot* opens a plot showing the time evolution of the diffKDE (e.g., see Figure 2 for example output). An alternative would be to incorporate this paragraph into section 3.1 and use your simulation example to highlight some of your programs capabilities.

36. Figure 5: Are you able to include the true curve as a reference?

37. Figure 6: Can you report the MISE or AMISE for all of the curves so there is some numerical reference on how they perform? Also, what about a third column using 1000 points or so to highlight where diffKDE does *really* well. It would be nice to see how the other methods perform when diffKDE effectively perfectly captures the truth (e.g., are Botev, Gaussian, Epan still way off when diffKDE is nearly perfect).

38. Figure 9: I would either connect the dots with a line or have no line at all. I think the best fit line is odd here.

**Technical Corrections**

General: Consider using simpler, more precise, descriptions of the concepts. There are various cases, some that are included in the following list, where your message gets lost due to word choice. Below is a list of the some of the technical and grammatical issues, but not all. I recommend the authors verify the manuscript is void of technical and grammatical issues before re-submission.

1. Line 2: Comma after geoscience.

2. Line 5: No need for a comma, "but incomplete because of the..."

3. Line 7: This sentence feels odd. Consider rewording. Something like, "To make diffusion-based KDE accessible for general use, we designed and developed..."

4. Line 8: "We demonstrate our tool on simulated and real marine biogeochemical data individually, and compare our results against other popular KDE method". Also, be clear on if the simulated data is marine biogeochemical data or is not related to marine data.

5. Line 11: This sentence reads awkwardly. Consider breaking it into two pieces. "The convergence ... smaller error. This is most notable for ... "

6. Line 12: I don't understand the use of "exemplify".

7. Line 13: Is this related to the real-world example? If so, perhaps move it to earlier in the paragraph.

8. Line 15: Comma after geoscience.

9. Line 18: "Such necessity is not only..."

10. Line 19: "... applications such as social science and financial or ecological research."

11. Line 21: "... by some distance or divergence measure between..."

12. Lines 33-43: Some of the sentences in this paragraph read awkwardly. For example, the first three feel like they could be combined to read more clearly. Consider reworking this paragraph.

13. Line 45: No comma after "is possible"

14. Line 53: "... choose between varying levels of smoothness by design."

15. Line 55: "... KDE with an accompanying Python package, *diffKDE* **cite or link package**."

16. Line 59: Remove ", so called"

17. Line 61: Starting with "Thus, ..." this sentence is confusion, consider rewording.

18. Line 72: You already defined PDFs, can remove "probability density functions".

19. Line 85: "integrated squared error (MISE) is ..."

20. Line 88: Define AMISE.

21. Line 117: "...Chaudhuri and Marron (2000) and its benefits were ..."

22. Line 119: What does "is" refer to in "... and is extended ..."

23. Line 123: No comma after (diffKDE)

24. Line 128: "When regarded as a PDE, the Dirac $\delta$-distribution puts all of the probability as the corresponding data point."

25. Line 131: This sentence feels out of place, consider incorporating it into a paragraph so it is not a stand alone sentence.

26. Line 133: Obliterates does not feel like the correct word.

27. Line 144: Both sentences start with "in".

28. Line 154: Unclear, consider rewording.

29. Line 171: What is Gaussian here?

30. Lines 178-179: Incorporate into the paragraph after it.

31. Line 199: "...division by $\boldsymbol{p}$ is applied column-wise."

32. Line 221: Is fineness the right word? Maybe size? Or resolution?

33. Figure 1: "The function $\Phi_h$ depends on the ..."

34. Line 228: Is this supposed to be a paragraph? Odd indenting.

35. Line 240: Starting with "It is built...", re-word this sentence.

36. Line 466: Chapter?

37. Line 474: Define the euphotic zone.

---

## Referee Comment (RC2)

In `A diffusion-based kernel density estimator (diffKDE, version 1) with optimal bandwidth approximation for the analysis of data in geoscience and ecological research,' Pelz et al. introduce a diffusion-based approach to kernel density estimation, as well as its implementation in Python. From what I understand, they treat input data points as delta function sources in a diffusion equation and run the diffusion calculation forward until a given stopping 'time'; they also allow for a user-specifiable diffusivity, that is a function of data space, allowing higher effective resolution in certain areas of the data space. They show that the KDE estimation method performs comparably to other KDE implementations in the Python scipy package, and they show that it tends to have lower mean-squared error for relatively small (O(100)) sample sizes.

This paper presents a novel statistical model, with the apparent innovations in the paper being twofold: (1) the use of a spatially-varying diffusivity in a diffusion-based KDE estimator, and (2) implementation of the method in an open-source language. With respect to the journal, GMD, the paper presents a new method for statistical modeling in general, with applicability to geoscience domains; so it seems to be within scope of the journal (though I have further comments on that below). Based on this, the paper seems publishable in principle within this journal. Overall, this seems like a cool technique!

In its current form, the paper has a number of issues that lead me to recommend a major revision: (1) the paper has widespread use of notation that is not well-explained and is not likely to be widely-known by a geoscience audience; (2) the literature review and discussion misses some key, current literature that is highly relevant; and (3) the current form of the paper appears to advance a statistical method and only briefly applies the technique to geoscientific data, making it unclear whether this paper is really within the scope of GMD. These issues are described in more detail below.

**Major comments**

**Widespread use of unclear notation**

One of the biggest issues with this paper is the widespread use of jargon and notation that is not common in the geosciences. For example, the paper makes widespread use of set notation (e.g., line 75, Equation 2), which is not commonly taught in geoscience curricula (I have no idea what the $\hat{f}\mathscr{R} \times \mathscr{R}\ldots$ notation in Equation 2 means, and I have quite a bit of training in math and statistics, including having published in statistics journals myself!). In another example, := notation is used (e.g., line 225), and I am not sure what it means here; I'm used to reading it as `is distributed as', but I don't think that's what is meant here (and what does the $= :$ mean in equation 30?). If notation like this is to be used, I suggest defining what the notation means at first use. My main concern here is that the notation may end up being a barrier to people reading this paper, which would then also inhibit this paper's potential impact.

Somewhat related to the above, there is ambiguous usage of the symbol $t$. In some places it is clear that it is meant as the bandwidth (e.g., line 113). In others (e.g., the left-hand side of Equation 10), it seems to mean 'time' in the sense of time-evolution of the diffused quantity $u$. I don't think that the bandwidth and the time are meant to be taken as being equivalent in this paper, so a symbol other than $t$ really should be used for the bandwidth. (Especially for this audience, where $t$ is almost always reserved to refer to time.) That said, I'm genuinely confused about Algorithm 1, where lines 4 and 5 of the algorithm seem to clearly be setting $T_p$ and $T_f$ as bandwidths, but line 7 seems to be treating $t$ as time and $T_p$ as a maximum time; so maybe bandwidth and time really are the same in this paper? If so, that needs to be stated *very* clearly and perhaps repeatedly, since that's quite unintuitive.

In the end, I'm not certain I understood the method well enough to review it thoroughly, which is a major concern.

**KDE literature review**

The introduction does a good job of describing KDE and the background of KDE, giving references to bandwidth choice up through Scott, 2012. However, there are some recent innovations in kernel density estimation that are highly relevant here:

- Bernacchia and Pigolloti (2011) derive a method for choosing both the kernel and its bandwidth in an `optimal' way. Note also that there are several geoscience applications of this approach, which can be found by looking at literature citing this paper.
- Davies and Baddeley (2017) describe a spatially adaptive bandwidth approach, which seems relevant here given the spatially-adaptive diffusivity used in this paper
- Chacón and Duong (2018) overview KDE methods and the variety of bandwidth selection methods used; this is especially relevant since Duong authored the widely used ks package in R, which would be a good, additional state-of-the-science package to compare against in Figures 6, 7, 9.
- Another relatively recent book overviews KDE methods: https://link.springer.com/book/10.1007/978-3-319-71688-6

I would also add that, for this audience, the introduction would be well-served by listing a number of examples of the use of KDE in geoscience literature.

**Clarifying how this manuscript is in scope for GMD**

My final concern relates to the scope. In my initial thinking, I considered recommending rejection of this paper because it seems like it might be out of scope for GMD. Most of the content and innovation in the paper relates to a general-purpose statistical method, so it almost seems like this might be more suitable for a statistics journal like JABES. I think that part of this impression comes across in the emphasis on the development of the statistical method itself, rather than its application (and/or potential application) in geosciences. I think this partly relates to the technical notation comment above too; because the notation does not seem typical of a geoscience paper, it kind of feels like it was written for a stats audience.

I spoke with GMD editors about this, and they indicated that they do think it is potentially in scope. But the paper should make a clear case for how this new method advances geoscientific modeling, and I think it might be good to emphasize applications a bit more than the method itself. I also think that the discussion of the method (i.e., section 2) could be revised to be much less rigorous (which would also help with the statistical notation comment); focus on making sure that a large portion of your potential geoscience audience can understand this method (and your innovation to it) rather than focusing on precise mathematical language. It also might help to modify the introduction by mentioning some specific uses of KDE in geoscience papers and also modifying the discussion to relate the advances in this paper back to those papers; what might have been different/better about the past papers if they had used your method?

I would also add that the paper should be more explicit about the innovations of this specific paper: in the abstract, the introduction, in the method section, and again in the conclusions. I do see that the innovation of this paper is discussed in lines 544-549, which is great, but the way this is written, it is not clear what specific aspects of this paper are new relative to the citations mentioned. It might help to revise Section 2 such that it is clear which equations are essentially 'background information' and which equations (or which parts of the equations) contain the innovation in this paper.

---

## Author Comment (AC1)

Dear Reviewer 1

We are thankful for the many helpful comments and suggestions for improving our manuscript. By addressing all of your points, we have made significant changes to the structure of our manuscript. We have also added another example that should be illustrative and inspiring for those readers who do not intend to fully understand all the technical details of the diffKDE. Your comments are shown in grey. Our responses are in black. We numbered comments and responses to allow cross-referencing. The line numbers are referring to the manuscript document with the tracked changes below.

**General Comments**

The authors present a novel kernel density estimation approach that combines a diffusion-based method with a pilot step to produce better estimates of the bandwidth - a classically difficult problem. The authors also develop a Python package, diffKDE, for easy use of their method and highlight some of the package's capabilities within the manuscript. diffKDE is benchmarked against pre-existing KDE methods using synthetic data and applied to real marine data, highlighting its applicability. However, the material is organized and presented in such a manner that detracts from its scientific contribution. Additionally, as noted in the points below, the verbiage, lack of technical definitions in multiple locations throughout the text, and sectioning of the ideas muddle the presentation of the material. I recommend the authors revise the manuscript and give special attention to the structure of the manuscript and the presentation clarity. I believe this work has potential and I look forward to the reading the revised manuscript.

**Specific Comments**

The organization of the paper detracts from the scientific contribution. Here are a couple examples:

**RC1 – 1** Section 2.1 gives background on the general kernel density estimator and the proposed diffusion-based kernel density estimator with pilot study. Then section 2.2 and 2.3 discuss discretization of the diff-KDE and implementation, respectively. This feels out of order. I would assume general background would be its own section (e.g., make section 2.1 its own section)

**RC1 – A1** We made section 2.1.1 its own section named "2 The general kernel density estimator" presenting the general introduction to KDEs.

**RC1 – 2** and the proposed method and all associated ideas would be grouped together (e.g., make sections 2.1.2 through 2.4 its own section, say section 3, and order these appropriately).

**RC1 – A2** We made the theoretical background of the diffusion KDE an own section named "3 The diffusion-based kernel density estimator" including the subsections for bandwidth selection and pilot estimation. The second and third paragraph of Section 3 are re-ordered and completed to first give a general introduction and motivation for the

diffusion-based KDE and explain how the bandwidth is related to the time propagation and then describe the connection to the Gaussian KDE (see RC2 – A8).

The following final parts of the Methods section are now restructured into two additional sections showing first the theory behind our new approach to the diffKDE in a section "4 The new bandwidth approximation and pilot estimation approach of the diffKDE" and second the explicit discretization and implementation of the diffKDE in "5 Discretization and implementation of the diffKDE". Redundant subheadings were removed.

We added the phrase: "...on an equidistant spatial grid $(x_i)_{n}^{i}=0 \in \bar{\Omega}$ of the spatial domain $\Omega \subseteq \mathbb{R}$." in l. 291.

We deleted the sentences: "The selected implementation is a straightforward approach using equidistant finite differences in space and time and a direct solution of the diffusion equation by an implicit Euler." and begin the new section 4 with "Our new approach solves the diffusion equation in three stages, ..."

We renamed the section describing the algorithm realized in Python to: "5.4 The diffKDE algorithm with optimized bandwidth" and the final section in this part, which describes the implemented functionality for direct visual output, to: "5.5 Pre-implemented functions for visual outputs"

**RC1 – 3** Regarding the order material is presented, there is no initial explanation as to why a pilot estimation is needed. It feels as though section 2.1.4 needs some prior context or motivation.

**RC1 – A3** We included an introduction for the need of pilot estimation into the chapter about bandwidth selection as: "This implicit dependency can be solved by so-called pilot estimation steps. Pilot estimates are generally rough estimates of f calculated in an initial step to use them for an approximation of Eq. 17 and 18, which later on serve to calculate a more precise estimate of f . A more detailed introduction into pilot estimation and its specific benefit for diffusion-based KDEs is presented in Sec. 3.2.

Botev et al. (2010) use an iterative scheme to solve the implicit dependency of the bandwidth parameter on the true distribution f. This additional…" in ll. 237 – 241.

**RC1 – 4** Section 3 contains the simulation example and real world example. I would makes these distinct sections (e.g., following the above point, sections 4 and 5). To this point, the first paragraph of section 3 is confusion. There is no clear distinction between the simulation explanation and the real world example. By separating these, that should mitigate any confusion.

**RC1 – A4** We separated the results section, focussing first on artificial data "6 Results on artificial data" and on real marine data "7 Results on marine biogeochemical data and outlook to model calibration".

In the introductory paragraph we deleted the sentence referring to different data sources and the specific mentioning of the results obtained by the pre-implemented functions ("Different data sources are chosen to best show possibilities and performance of the diffKDE. Additionally, snapshots of the pre-implemented plot routines are given as examples." ll. 507 – 509).

We moved the final paragraph of the prior Results introduction to the beginning of the new second results section "Results on marine biogeochemical data and outlook to model calibration" and rephrased it to: "The performance of the diffKDE is now illustrated with real data of a) measurements of carbon isotopes (Verwega et al., 2021, Verwega et al., 2021}), b) of plankton size (equivalent spherical diameter) (Lampe et al., 2021) and c) remote sensing data (Sathyendranath et al., 2019, 2021). We chose these data because we propose to apply the diffKDE for the analysis of field data for assessment and optimization of marine biogeochemical- as well as size-based ecosystem models. The carbon isotope data have been collected to constrain model parameter values of a marine biogeochemical model that incorporates this tracer as a prognostic variable (Schmittner and Somes, 2016." in ll.634 – 641.

Furthermore, we deleted the sentences: "In this final part, we show the diffKDE's performance on real marine biogeochemical field data. We chose two example data: A set of $\delta^{13}C$ in particulate organic carbon (POC) (Verwega et al., 2021) data and a set of plankton size spectra data (Lampe et al., 2021). " to avoid redundancies.

We incorporated two subsections into "7 Results on marine biogeochemical data and outlook to model calibration" namely "7.1 Performance analyses on organic carbon-13 isotope data" and "7.2 Performance analyses on plankton size spectra data"

I strongly recommend reorganizing the paper by grouping similar ideas by sections with appropriate sub-sections.

**RC1 – 5** Some of the verbiage and notation used in this paper may preclude the general audience of GMD from understanding its scientific merit. There are some examples included in the list below, but I recommend the authors check the manuscript and make sure all mathematical terms are defined and avoid using overly technical notation if possible.

**RC1 – A5** We clarified and simplified the usage of mathematical terminology at various places in detail listed in the answer to RC2 (see RC2 – A1, RC2 – A4, RC2 – A5, RC2 – A6, RC2 – A7).

Abstract: Some of the sentences need some work. For example:

**RC1 – 6** Line 1: A PDF is a function defining the probability of a random variable taking on a specific value. I don't like the use of observed or simulated variables or the phrase, "comprise basic information", as I think this is a little too weak of wording.

**RC1 – A6** We rephrased the sentence to: "Probability density functions (PDFs) provide information about the probability of a random variable taking on a specific value." in ll. 1-2.

**RC1 – 7** Line 6: Starting with "A diffusion-based KDE..." Why is this the case? Maybe rephrase to say, "Diffusion-based KDEs have been shown to provide a useful approach ..."

**RC1 – A7** We deleted this sentences and formulated more precisely: "In this study, we designed and developed a new implementation of a diffusion-based KDE as an open source Python tool to make diffusion-based KDE accessible for general use. Our new diffusion-based KDE provides (1) consistency at the boundaries, (2) better resolution of multimodal data, (3) and a family of KDEs with different smoothing intensities. We

demonstrate our tool on artificial data with multiple and boundary close modes and on real marine biogeochemical data, and compare our results against other popular KDE methods." in ll. 8 – 13.

**RC1 – 8** Line 10: What is boundary close data and what details are suppressed? Do you mean your approach produces a smooth surface that is robust to noise and outliers?

**RC1 – A8** We rephrased the sentence to: "Our estimator is able to detect relevant multiple modes and it resolves modes that are located closely to a boundary of the observed data interval. Furthermore, our approach produces a smooth graph that is robust to noise and outliers." in ll. 14 – 17.

**RC1 – 9** Also, there are various other places in the manuscript that use "details", but it is unclear what is being referred to.

**RC1 – A9** By details we refer to multiple modes that can be resolved with the diffKDE. We used the term "details" in the example of the plankton size spectra. Since we have now included in the introduction section a description of the relevance and the role of KDE for analyzing continuous size spectra, we explain what we mean by "details": "The identification of structural details in the size spectra, such as distinct elevations (modes) and troughs within certain size ranges, is useful, since they can reveal some of the underlying structure of the plankton foodweb." in 52 – 53.

In addition, we wrote in subsection (7.2 Performance analyses on plankton size spectra): "This procedure avoided over-fitting but was also prone to over-smoothing, which can mask details, such as troughs in specific size ranges." in ll. 679 – 680.

**RC1 – 10** The introduction falls a short. There is a nice body of literature on diffusion-based KDE and it would be beneficial to this manuscript if it is put into context with the current literature.

**RC1 – A10** We added the reference

Majdara, A., & Nooshabadi, S. (2019). Nonparametric density estimation using copula transform, bayesian sequential partitioning, and diffusion-based kernel estimator. *IEEE Transactions on Knowledge and Data Engineering*, *32*(4), 821-826.

as: "..., especially for the resolution of multiple modes (e.g. Majdara and Nooshabadi, 20)." in l 89.

And

Li, G., Lu, W., Bian, J., Qin, F., & Wu, J. (2019). Probabilistic optimal power flow calculation method based on adaptive diffusion kernel density estimation. *Frontiers in Energy Research*, *7*, 128.

Santhosh, D., & Srinivas, V. V. (2013). Bivariate frequency analysis of floods using a diffusion based kernel density estimator. *Water Resources Research*, *49*(12), 8328-8343.

Xu, X., Yan, Z., & Xu, S. (2015). Estimating wind speed probability distribution by diffusion-based kernel density method. *Electric Power Systems Research*, *121*, 28-37.

as: "The improved structure resolution has for example already shown useful for the optimization of photovoltaic power er generation (Li et al., 2019), analysis of flood frequencies (Santhosh and Srinivas, 2013) or the prediction of wind speed (Xu et al., 20)." in 90 – 91.

**RC1 – 11** Line 19: Any citations to support the claim for the need in other model applications? This seems like a great place to include more references.

**RC1 – A11** We have thoroughly revised the first paragraphs of the introduction section, and have included references to studies where the computation and analyses of PDFs seem particularly useful (Dessai et al., 2005; Perkins et al., 2007; Palmer 2012). We also added: "Obtaining high quality approximations of nonparametric PDFs is certainly not limited to applications in the geosciences but is likely desirable in other scientific fields as well. In aquatic ecological research, for example, continuous plankton size spectra can be well derived from PDFs of cell size measurements sorted by individual species or plankton groups (Quintana et al., 2008; Schartau et al., 2010;  Lampe et al., 2021). The identification of structural details in the size spectra, such as distinct elevations (modes) and troughs within certain size ranges, is useful, since they can reveal some of the underlying structure of the plankton foodweb. A typical limitation of the approach described in Schartau et al. (2010) and Lampe et al. (2021) is the specification of an estimator for the continuous size spectra, such that all significant details are well resolved." in ll. 48 – 55.

**RC1 – 12** Lines 24-27: Your definition of a PDF does not seem correct. You need to explicitly define each element of you probability space $\Omega$, A, P . Also, I assume A is a σ-algebra and $\Omega$ is the sample space. In which case I think you mean to have $f : \Omega \to [0,\infty)$ (also, open bracket on the right).

**RC1 – A12** We simplified the definition of the PDF to the essential parts and re-wrote it to: "Mathematically formulated, PDFs are integrable non-negative functions $f : \Omega \to [0, \infty)$ from a sample space $\Omega \subseteq \mathbb{R}$ into the non-negative real numbers with $\int_{-\infty}^{\infty} f(x)\, dx = 1$" in l. 56.

**RC1 – 13** Also, the sentence, "By definition, ..." should be reworded.

**RC1 – A13** We re-phrased the sentence to: "PDFs correspond to the probability $\mathbb{P}$ of the occurrence of a data value $X \in \mathbb{R}$ within a specific range $[a, b] \subseteq \mathbb{R}$ via the relationship…" in ll. 59 – 60.

**RC1 – 14** Line 41: I like the list, but are there any more current (last 5 or so years) papers that could be included? A cursory literature search returned quite a few recent papers on the topic.

**RC1 – A14** See RC2 – A10.

**RC1 – 15** Line 48: Define δ-distribution, or say something like, "which will be formally defined in section xxx.".

**RC1 – A15** We changed the sentence accordingly and added directly following the definition of the diffKDE: "In general, δ is defined by δ(x)=0 for all xR \{0} and $\int_{-\infty}^{\infty}$δ(x)=1 (Dirac, 1927)." in ll. 207 – 208.

**RC1 – 16** Line 75: Are all $X_j$ assumed to come from the same distribution or are they completely independent? Should they be independent or independent and identically distributed?

**RC1 – A16** For the general mathematical formulation we need them to be independent and identically distributed. We changed the sentence to: "...independent identically distributed real random variables." in ll. 118 – 119.

**RC1 – 17** Line 78: Is n the same as N? If so, verify the use of n and N throughout the paper.

**RC1 – A17** n will be later on the number of spatial discretization points of the spatial domain $\Omega$. N is the number of data pints $X_i$. We corrected the notation in this formula. As well as in l. 438 in the second paragraph of section 5.4.

**RC1 – 18** Line 78 and generally: This is a personal choice, but I feel your paper would be more accessible to the general GMD audience if you change your function notation. Consider

$f(x; h) = 1/nh \, \Sigma K((x-X_j)/h)$ , where $f : R \times R_{>0} \rightarrow R_{\geq 0}$.

**RC1 – A18** We changed the function notation in agreement with RC2 to:

$f(x; h) = 1/nh \, \Sigma K((x-X_j)/h)$ , where $x\epsilon R$, $h\epsilon R_{>0}$ and $f(x; h)\epsilon R_{\geq 0}$

Furthermore, we changed Eq. 4

$MISE(\hat{f})(h) = E\int R(\hat{f}(x; t) - f(x))^2 dx$, where $h \epsilon R_{>0}$

and Eq. 11 to

$\Phi_h(x) = \dots$ , , where $x, \Phi_h(x) \epsilon R$

**RC1 – 19** Line 86: Should it be $R_{>0}$ or $R_{\geq 0}$?

**RC1 – A19** It is $R_{>0}$, because the bandwidth shall always be non-negative.

**RC1 – 20** Line 91: It is not clear where equation (6) comes from. It would be helpful if this result, or derivation, is included in the appendix.

**RC1 – A20** We changed this part to include a more precise description of the prerequisites necessary for this result as well as a more accessible formulation of it as: "A kernel function K that suffices the additional conditions

$\int_R yK(y) \, dy = 0$, $\int_R y^2|K(y)| \, dy < \infty$, $\int_R y^2 K(y) \, dy = k_2 \epsilon R \setminus \{0\}$,

is a second order kernel as its second moment $\int_R y^2 K(y) \, dy$ is its first non-zero moment. Those kernels are positive and together with the final condition from Eq. 3 they are PDFs themselves. For the general KDE from Eq. 2 with a second order kernel the optimal bandwidth can be calculated as…" in ll. 141 – 146.

Furthermore, we added in Appendix A a more detailed explanation of the connection between the original idea by Parzen (1962) and our specific application, as well as a source more precisely describing the derivation of this result (Silverman, 1986): "The derivation of the optimal bandwidth choice for a KDE was already described in Parzen

(1962) and can be found in more detail in Silverman (1986). The additional conditions stated in Eq. 7 to the kernel function

$$\int_R yK(y)dy = 0 \text{ and } \int_R y^2 K(y)dy = k_2 \epsilon R \backslash \{0\}$$

correspond to the order of the kernel being equal to 2 (Berlinet, 1993). For such kernels Silverman (1986) showed the minimizer of the asymptotic mean integrated squared error to be

h = ...

In our context of working with the squared bandwidth $t=h^2$ this optimal bandwidth choice becomes $t = ( \|K\|_{L^2}^2 / (N k_2^2 \|f''\|_{L^2}^2))^{2/5}$", which equals Eq. 13."

**RC1 – A21** We changed the function notations in agreement with RC2 to:

$K_E(w) = \frac{3}{4}(1 - w^2)$ , where $w \in R$ and $K_E(w) \in R_{\geq 0}$.

and

$\Phi(w) = 1/\sqrt{2\pi} \ e^{-1/2 \ w^2}$, where $w \in R$ and $\Phi(w) \in R_{\geq 0}$.

**RC1 – 22** Line 109: Is the diffusion based approach an alternative method? Better? Also, different approach in what sense? Make this more clear.

**RC1 – A22** We expanded this sentence to become an introductory paragraph: "The diffusion-based KDE provides a different approach to Eq. 2 by solving a partial differential equation instead of the summation of kernel functions. This different calculation offers three main advantages: (1) consistency at the boundaries can be ensured by adding Neumann boundary conditions (2) better resolution of multimodal data can be achieved by the inclusion of a suitable parameter function in the differential equation leading to adaptive smoothing intensity (3) a family of KDEs with different bandwidths is produced as a by-product of the numerical solution of the partial differential equation in individual time steps." in ll. 167 – 172.

**RC1 – 23** Line 110: What is "It"? The KDE? Also, "progresses up to an estimate at the final time," is unclear. What progresses and what estimate?

**RC1 – A23** We changed the sentence to: "This KDE solves the partial differential equation describing the diffusion heat process, starting from an initial value based on the input data $(X\_j)_{j=1}^N$ and progresses forward in time to a final solution at a fixed time $T\epsilon R_{>0}$." in 173 – 175.

**RC1 – 24** Line 113: see comment 11.

**RC1 – A24** We changed the function notation in agreement with RC2 to:

, where $x \ \epsilon \ R$, $t \ \epsilon \ R>0$ and $\Phi(x; t) \ \epsilon \ R_{\geq 0}$

**RC1 – 25** Line 121: What is the importance of acting inversely proportional to the diffusion quotient? Expand on this point.

**RC1 – A25** We added the sentences: "This parameter function allows to influence the intensity of the diffusion applied adaptively depending on the location $x \epsilon R$. Its role and specific choice is discussed in detail in Sec. 4." in ll. 197 – 198.

RC1 – 26 Line 136: How do you set p properly? It is not clear at this point. (E.g., Line 166 starting with "Choosing p to be ..." could be incorporated here)

**RC1 – A26** We changed the sentence to: "Thus, choosing p to be a function allows for a spatially dependent influence on the smoothing intensity, which solves the prior problem of…" in ll. 217 – 219.

RC1 – 27 Line 147: Similar to comment 13, it is not clear where this equation comes from, consider adding its derivation to the appendix. Also, define all the terms in the equation (e.g., ... where □ · □L2 is the L2-norm...).

**RC1 – A27** This equation is a direct citation from Botev et al. (2010). We added the sentences: "…, where $|| \cdot ||_{L2}$ is the L2-norm and $\mathbf{E}( \cdot )$ the expected value. The proof of this equation is in detail given in Botev et al. (2010)." in ll. 232 – 233.

RC1 – 28 Line 153: How or why is the additional effort avoided in your approach?

**RC1 – A28** We extended the sentence to: "This additional effort is avoided in our approach by directly approximating f with a simple data-based bandwidth approximation in detail described in Sec. 4." in ll. 242 – 243.

RC1 – 29 Line 168: What is "resolves the unexpected structure"

**RC1 – A29** We rephrased the sentence to: "Low smoothing resolves more variability within areas with many similar values (high density), while the intensity of smoothing is increased where data values are more dispersed."

RC1 – 30 Line 185: The second equal should be an approximate sign. Same with multiple places following. Eq 15, 16, ...

**RC1 – A30** We changed the notation in l. 321**,** Eq. 21, 22, 23, 24, 26, 27, 28 and 30 and rephrased the final sentence of this section to: "By these calculations the solution of the partial differential equation from Eq. 14 can also be approximated by solving the system of ordinary differential equations:" in ll. 339 – 340.

RC1 – 31 Line 201: Bold u?

**RC1 – A31** We changed the font of *u* to bold.

RC1 – 32 Line 208: Why not an explicit? Does the solver have stability issues? Is this relevant?

**RC1 – A32** The explicit Euler is not A-stable, which is the implicit. This is why we chose the implicit approach to guarantee convergence of the KDE. We added the sentence: "The implicit Euler method is chosen at this place, since it is A-stable and by this ensures convergence of the solver." and rephrased the following to: "Eq. 33 together with the initial value Eq. 16 describes an…" in ll. 351 – 353.

RC1 – 33 Line 220: Define Bρ(0).

**RC1 – A33** We added: ", where $B_\rho(0) = \{x \in R; |x - 0| < \rho\} = (-\rho, \rho)$ is the open subset of R centered around R with radius $\rho$." in l. 363.

**RC1 – 34** Line 222: How is $|\Omega|/n = h$ calculated? Moreover, isn't $\Omega = R$?

**RC1 – A34** In general, $\Omega$ is a real subset of R, since we are only calculating over a finite data domain. The spatial stepsize h is the length of the domain $\Omega$ divided by the number of spatial discretization points. We rephrased the respective sentence to: "The spatial discretization step size $h \in R_{>0}$ equals the length of the domain $|\Omega|$ divided by the number of spatial discretization points n, namely $h = |\Omega|/n$. This relationship provides the dependency…" in ll. 365 – 366.

$\Phi_h$ shall be defined on R instead of $\Omega$ to make the integration possible (only its support lies in $\Omega$). We corrected the definition of $\Phi_h$ in Eq. 36.

**RC1 – A35** We added the definition and rephrased the sentence to: "Then $\Phi_h$ is non-negative for all $h \in R_{>0}$ and as a composition of integrable functions integrable with $\int \Phi_h(x)\, dx = 1$ (see App. B) and $\Phi_h \in L^1(R) = \{f : R \to R; f$ integrable and $\int |f(x)|\, dx < \infty\}$." in ll. 375 – 376.

**RC1 – 36** Line 259: Define iqr. A non-statistical audience may not know what this is.

**RC1 – A36** We added a definition os iqr as: "The iqr is the interquartile range defined as $iqr\,(data) = q\,(0.75) - q\,(0.25)$. The value $q\,(0.25)$ denotes the lower quartile and describes the value in data, at which $25\%$ of the elements in data have a value smaller than $q\,(0.25)$. $q\,(0.75)$ denotes the upper quartile and describes the analogue value for $75\%$ (Dekking et al., 2005)." in ll. 297 – 299.

**RC1 – 37** Line 271: Is the discretization of $\Omega$ the same as $\Omega$? See line 180. Also, if this is the case, there are multiple instances throughout the paper where this appears.

**RC1 – A37** We changed this part to: " ...on the discretization $(x_i)_{i=0}^n$ of $\Omega$ as…." in l. 303.

Furthermore, we changed a similar instance in l. 433 in the description of the algorithm to be now: "The spatial grid discretizing $\Omega$ is setup according…"

**RC1 – 38** Line 292: What do you mean by the variable $\Omega$? Is $\Omega$ the spatial domain or something else?

**RC1 – A38** We rephrased the sentence to: "This leads to a full set of $n + 1$ equidistant discretization points $(x_i)_{i=0}^n$ saved in a vector variable denoted in Alg. 1 as $\Omega$. The spatial discretization $(x_i)_{i=0}^n$ includes…" in ll. 440 – 441.

**RC1 – 39** Table 1: Why is the default number of spatial discretization intervals 1004 and not some value dependent on the dimension of $\Omega$ or the concentration and number of observed points?

**RC1 – A39** We chose this value to be comparable to the implementation by Hennig (2021) after Botev et. al (2010), where the default value was also set to be n=1024 but also possible to be set by the user.

**RC1 – A40** We changed the sentence to: "The temporal solutions are calculated *timesteps* ∈ N times in equidistantly increasing time steps until…" in ll. 459 - 460.

**RC1 – A41** We rephrased this part to: "The times are the 20 timesteps used for the calculation of u as defined in Tab. 1 followed by additional 10 up to the doubled approximated optimal final iteration time 2 $T^\square$. The time step size for the solutions between $T^\square$ and $T^\square$ are doubled because of the smaller changes in the solution for larger times as for example visible in Fig. 5." in ll. 475 – 478.

**RC1 – 42** Section 2.4: I think this section could use a major rework to make it more readable. Also, there are figures that relate to each of these functions. Why not relate the function to its corresponding figure? For example, "The function call evol plot opens a plot showing the time evolution of the diffKDE (e.g., see Figure 2 for example output). An alternative would be to incorporate this paragraph into section 3.1 and use your simulation example to highlight some of your programs capabilities.

**RC1 – A42** We linked the corresponding Figures to the respective function introduction as: "The function call *evol_plot* opens a plot showing the time evolution of the diffKDE (e.g., see Figure 2 for example output)." in l. 489.

and "The function call *pilot_plo*t opens that shows the diffKDE together with its pilot estimate p, showing the intensity of local smoothing (e.g., see Fig. 3 for example output)." in ll. 495 – 496.

and "The function call *custom_plot* opens an interactive plot, allowing the user to slide through different approximation stages of the diffKDE (e.g., see Fig. 4 for example output)." in ll. 498 – 499.

Furthermore, we rephrased ll. 490 – 491 to: "In the background the initial values are drawn. The y-axis range is cut off at 20 %…"

and l. 493 to: "The diffKDE is drawn in a bold blue line."

and l. 493 to: "….provides the user with insight…"

and l. 500 to: "At the bottom of this plot…"

and deleted the part: ", the intensity of smoothing at different localizations." in ll. 503 – 504.

**RC1 – 43** Figure 5: Are you able to include the true curve as a reference?

**RC1 – A43** We included the shaded distribution in the background in consistency with the other graphics generated from known distributions.

**RC1 – 44** Figure 6: Can you report the MISE or AMISE for all of the curves so there is some numerical reference on how they perform?

**RC1 – A44** The insight into the numerical performance of the four KDEs is presented in Figure 9 and Table C1 by the error between the estimates and the true distribution of the first row of Figure 6 for different sample sizes measured by the Wasserstein distance.

We added the MISE (approximate by 100 different random samples of the respective sample sizes) as a second panel to Fig. 9. and expanded the figure caption to: "(a) shows

the error calculated with the Wasserstein distance and (b) with the MISE. The MISE is calculated after Eq. 2 from 100 different random samples."

Furthermore, we added the MISE values from Fig. 9 to Table C1 and the sentence: "...and the MISE defined in Eq. 4. For the approximation of the expected value in Eq. 4 we applied an averaging of the integral value for 100 different random samples for each observed sample size." in ll. 605 – 606.

RC1 – 45 Also, what about a third column using 1000 points or so to highlight where diffKDE does really well. It would be nice to see how the other methods perform when diffKDE effectively perfectly captures the truth (e.g., are Botev, Gaussian, Epan still way off when diffKDE is nearly perfect).

**RC1 – A45** We added a third column to the figure with 1000 data points and moved the integral values to a table to ensure readability of the graphics. We adapted the caption to: "The plots (a), (b) and (c) show KDEs of random samples of the trimodal distribution defined in Eq. 39, (d), (e) and (f) the same for a lognormal distribution. The left figure column is constructed from 50 random samples, the middle from 100 and the right from 1000."

Furthermore, we adapted the describing part of the Figure in the corresponding section to: "The random samples are 50, 100 and 1000 data points of each distribution…" in ll. 566 – 567.

And added: "In the 1000 random samples test the diffKDE best detects the left mode and the Botev KDE the two others best. Generally, diffKDE and Botev KDE are closely aligned in this case. As well are Gaussian and Epanechnikov KDE are closely aligned, but with a worse fit of all structures of the true distribution." in ll. 575 – 578.

As well as: "In the 1000 random samples test with the lognormal distribution are again diffKDE and Botev KDE closely aligned as well as Gaussian and Epanechnikov KDE. The first two are very close to the true distribution, but resolve too much structure of the random sample. The diffKDE resolves more structure in the area close to 0 and becomes smoother towards the tail of the distribution. The Botev KDE performs the other way around and provides a smoother estimate close to 0 and more structure of the random sample towards higher data values." in ll. 580 – 585.

Finally, we rephrased the last sentence of this paragraph to: "An analysis of the integral of the KDEs over the observed domain is presented in Tab. 3 and reveals…" in l. 585.

The table also includes the integral values from Figure 7 for consistency, which are now also removed fro the graphics for better readability. We rephrased the last sentence of the respective paragraph to: "The integrals of the KDEs are also presented in Tab. 3 and our implementation is again always exactly 1." in ll. 594 – 595.

RC1 – 46 Figure 9: I would either connect the dots with a line or have no line at all. I think the best fit line is odd here.

**RC1 – A46** We re-did the figure without the regression, but connected the dots and added a grid. We deleted the sentences in the caption of Figure 9 and its description in the respective section that referred to the regression line.

We moved Table 3 to the appendix and referred to the corresponding KDEs in Figure 9 in the table caption as: "Error convergence ob the observed KDEs in Fig. 9."

Technical Corrections

**RC1 – 47** General: Consider using simpler, more precise, descriptions of the concepts. There are various cases, some that are included in the following list, where your message gets lost due to word choice. Below is a list of the some of the technical and grammatical issues, but not all. I recommend the authors verify the manuscript is void of technical and grammatical issues before re-submission.

**RC1 – A47** We added more simple and concise descriptions to numerous parts in all sections of the manuscript as described (see tracked changes document), in addition to the here listed points in RC2 – A1, RC2 – A4, RC2 – A5, RC2 – A6, RC2 – A7, RC2 – A8, RC2 – A9.

**RC1 – 48** Line 2: Comma after geoscience.

**RC1 – A48** We changed the sentence to: "In geoscience, data …" in l. 2.

**RC1 – 49** Line 5: No need for a comma, "but incomplete because of the..."

**RC1 – A49** We changed the sentence to: "Existing KDEs are valuable but problematic because…" in ll. 5 -6.

**RC1 – 50** Line 7: This sentence feels odd. Consider rewording. Something like, "To make diffusion-based KDE accessible for general use, we designed and developed..."

**RC1 – A50** We changed the sentence to: "In this study, we designed and developed a new implementation of a diffusion-based KDE as an open source Python tool to make diffusion-based KDE accessible for general use. Our new diffusion-based KDE provides…" in ll. 8-10.

**RC1 – 51** Line 8: "We demonstrate our tool on simulated and real marine biogeochemical data individually, and compare our results against other popular KDE method". Also, be clear on if the simulated data is marine biogeochemical data or is not related to marine data.

**RC1 – A51** We used artificial data from known distributions as main test cases, since these provide a direct comparison of the KDEs to the true distribution. Furthermore, we used different marine biogeochemical data to provide some real-world application examples. To clarify this, we changed the sentence to: "We demonstrate our tool on artificial data with multiple and boundary close modes and on real marine biogeochemical data, and compare our results against other popular KDE methods." in ll. 11 – 13.

**RC1 – 52** Line 11: This sentence reads awkwardly. Consider breaking it into two pieces. "The convergence ... smaller error. This is most notable for ... "

**RC1 – A52** We broke the sentence down accordingly to: "..., but with a generally smaller error. This is most notable for…" in ll. 17 – 18.

**RC1 – 53** Line 12: I don't understand the use of "exemplify".

**RC1 – A53** We changed the sentence to: "We discuss the general applicability of such KDEs for data-model comparison in geoscience.." in ll. 18 – 19.

**RC1 – 54** Line 13: Is this related to the real-world example? If so, perhaps move it to earlier in the paragraph.

**RC1 – A54** Yes, this is one of the real-world examples. We shifted the sentence to ll. 13 – 14.

**RC1 – 55** Line 15: Comma after geoscience.

**RC1 – A55** We changed the sentence to: "In geoscience, the application of…" in l. 24.

**RC1 – 56** Line 18: "Such necessity is not only..."

**RC1 – A56** We rephrased the sentence accordingly to: ". Accordingly, there is a strong demand for the analysis of model simulations on various temporal and spatial scales and to evaluate these results against observational data." in ll. 27 – 29.

**RC1 – 57** Line 19: "... applications such as social science and financial or ecological research."

**RC1 – A57** See RC1 – A56.

**RC1 – 58** Line 21: "... by some distance or divergence measure between…"

**RC1 – A58** We changed the sentence accordingly in ll.42 – 43.

**RC1 – 59** Lines 33-43: Some of the sentences in this paragraph read awkwardly. For example, the first three feel like they could be combined to read more clearly. Consider reworking this paragraph.

**RC1 – A59** We changed the beginning of the paragraph to: "A KDE is based on a kernel function and a smoothing parameter. The kernel function is ideally chosen to be a PDF itself, usually unimodal and centered around zero (Sheather, 2004). The estimation process sums up the kernel function sequentially centered around each data point." in ll. 67 – 69.

and later on in the paragraph: "If it is larger, more structure becomes smoothed out (Jones et al., 1996), and information from single data points can get lost." in ll. 72 – 73.

**RC1 – 60** Line 45: No comma after "is possible"

**RC1 – A60** We changed the sentence to: "This perspective change is possible because the Gaussian …" in l. 83.

**RC1 – 61** Line 53: "... choose between varying levels of smoothness by design."

**RC1 – A61** We changed the sentence accordingly in ll. 94 – 95.

**RC1 – 62** Line 55: "... KDE with an accompanying Python package, diffKDE cite or link package."

**RC1 – A62** We changed the sentence to: "In this study, we present a new, modified diffusion-based KDE with an accompanying Python package, diffKDE (Pelz and Slawig, 2023)." in ll. 96 – 97.

**RC1 – 63** Line 59: Remove ", so called"

**RC1 – A63** We changed the sentence accordingly in l. 101.

**RC1 – 64** Line 61: Starting with "Thus, ..." this sentence is confusion, consider rewording.

**RC1 – A64** We rephrased the sentence to: "This allows for an interactive investigation of estimated densities at different smoothing intensities." in ll. 103 – 104.

**RC1 – 65** Line 72: You already defined PDFs, can remove "probability density functions".

**RC1 – A65** We changed the sentence accordingly in l. 114.

**RC1 – 66** Line 85: "integrated squared error (MISE) is ..."

**RC1 – A66** We changed the sentence accordingly in l. 133.

**RC1 – 67** Line 88: Define AMISE.

**RC1 – A67** We added the definition as: "In the following, we will work with the asymptotic MISE denoted as AMISE, which describes the asymptotic behavior of the MISE for the bandwidth parameter approaching zero $h \to 0$, meaning $\lim_{t \to 0} \mathrm{MISE}(f)(t) \,/\, \mathrm{AMISE}(f)(t) = 1$." in ll. 135 – 136.

**RC1 – 68** Line 117: "...Chaudhuri and Marron (2000) and its benefits were ..."

**RC1 – A68** We connected the sentences as suggested in ll. 193 – 194.

**RC1 – 69** Line 119: What does "is" refer to in "... and is extended ..."

**RC1 – A69** We rephrased the sentence to: "Our implementation of the diffusion KDE is based on Chaudhuri and Marron (2000), which we extended by some advancements proposed by Botev et al. (2010):" in ll. 195 – 196.

**RC1 – 70** Line 123: No comma after (diffKDE)

**RC1 – A70** We changed the sentence accordingly in l. 200.

**RC1 – 71** Line 128: "When regarded as a PDE, the Dirac δ-distribution puts all of the probability as the corresponding data point."

**RC1 – A71** We changed the sentence accordingly in ll. 208 – 209.

**RC1 – 72** Line 131: This sentence feels out of place, consider incorporating it into a paragraph so it is not a stand alone sentence.

**RC1 – A72** We moved the sentence to l. 202 directly following the definition of the diffKDE.

**RC1 – 73** Line 133: Obliterates does not feel like the correct word.

**RC1 – A73** We changed the sentence to: "). This makes the identification of one single optimal bandwidth unnecessary, which is ideal because the optimal value can be specific to a certain application and is often debated (….)." in ll. 213 – 215.

**RC1 – 74** Line 144: Both sentences start with "in".

**RC1 – A74** We changed the first sentence to: "We stressed Eq. 13 that…" in l. 228.

**RC1 – 75** Line 154: Unclear, consider rewording.

**RC1 – A75** We rephrased the sentence to: "The possible difficulties in finding one single optimal bandwidth (e.g. Scott, 2012) do not arise in the calculation of the diffKDE, by default." in ll. 244 – 246.

RC1 – 76 Line 171: What is Gaussian here?

**RC1 – A76** We rephrased the sentence to: "This approach combines Gaussian KDE and diffKDE interchangeably…" in ll. 264 – 265.

RC1 – 77 Lines 178-179: Incorporate into the paragraph after it.

**RC1 – A77** We merged the paragraphs as suggested in ll. 311 – 121 (not visibly resolved in the tracked changes document).

RC1 – 78 Line 199: "...division by p is applied column-wise."

**RC1 – A78** We changed the sentence accordingly in ll. 335 – 336.

RC1 – 79 Line 221: Is fineness the right word? Maybe size? Or resolution?

**RC1 – A79** We changed the phrase to: "step size" in l. 364.

RC1 – 80 Figure 1: "The function $\Phi_h$ depends on the ..."

**RC1 – A80** We changed the sentence accordingly in the caption.

RC1 – 81 Line 228: Is this supposed to be a paragraph? Odd indenting.

**RC1 – A81** The new paragraph begins with "Now, let …". We corrected the indenting accordingly in l. 374.

RC1 – 82 Line 240: Starting with "It is built...", re-word this sentence.

**RC1 – A82** We rephrased the sentence to: "Our new approach solves the diffusion equation in three stages, where the first two provide pilot estimation steps for the diffKDE." in ll. 268 – 269.

RC1 – 83 Line 466: Chapter?

**RC1 – A83** We changed the sentence to: "Since we have already done this in Sec. 6.2, we…" in ll. 644 – 645.

RC1 – 84 Line 474: Define the euphotic zone.

**RC1 – A84** We added the following sentences for clarification: "The euphotic zone describes the uppermost ocean layer that is attenuated with light to enable photosynthesis that produces organic matter (Kirk, 2011). While its depth can vary in nature (Urtizberea et al., 2013), here we pragmatically selected included data in the upper 130 m consistent with the analysis in the data set description (Verwega et al., 2021)." in 654 – 657.

Dear Reviewer 2

We greatly appreciate your constructive and helpful comments. We have addressed all your points and feel that our revised version of the manuscript is a significant improvement. Original comments and questions are shown in grey, while our responses are colored black in the text below. All points and our responses are numbered to allow cross-referencing. The line numbers are referring to the manuscript document with the tracked changes below.

In `A diffusion-based kernel density estimator (diffKDE, version 1) with optimal bandwidth approximation for the analysis of data in geoscience and ecological research,' Pelz et al. introduce a diffusion-based approach to kernel density estimation, as well as its implementation in Python. From what I understand, they treat input data points as delta function sources in a diffusion equation and run the diffusion calculation forward until a given stopping 'time'; they also allow for a user-specifiable diffusivity, that is a function of data space, allowing higher effective resolution in certain areas of the data space. They show that the KDE estimation method performs comparably to other KDE implementations in the Python scipy package, and they show that it tends to have lower mean-squared error for relatively small (O(100)) sample sizes.

This paper presents a novel statistical model, with the apparent innovations in the paper being twofold: (1) the use of a spatially-varying diffusivity in a diffusion-based KDE estimator, and (2) implementation of the method in an open-source language. With respect to the journal, GMD, the paper presents a new method for statistical modeling in general, with applicability to geoscience domains; so it seems to be within scope of the journal (though I have further comments on that below). Based on this, the paper seems publishable in principle within this journal. Overall, this seems like a cool technique!

In its current form, the paper has a number of issues that lead me to recommend a major revision:

**RC2 – 1** the paper has widespread use of notation that is not well-explained and is not likely to be widely-known by a geoscience audience;

**RC2 – A1** In addition to the specifically mentioned points below and those changes made according to RC1, we also added a more detailed description of the general conditions for a KDE as: "… meaning that K is bounded, integrable, and for the limit $y \to \infty$ decreases faster to zero than y approaches infinity. The final condition means that K integrates to 1 over the whole real domain, which implies that also the KDE $\hat{f}$ integrates to 1 as it is necessary for a PDF." in ll. 123 – 125.

**RC2 – 2** the literature review and discussion misses some key, current literature that is highly relevant; and

**RC2 – A2** See RC2 – A10 and RC1 – A10 for a detailed response.

**RC2 – 3** the current form of the paper appears to advance a statistical method and only briefly applies the technique to geoscientific data, making it unclear whether this paper is really within the scope of GMD. These issues are described in more detail below.

**RC2 – A3** We understand the reviewers comment and have therefore decided to include another example, in which we apply the technique to some regional ocean remote sensing (satellite) data of chlorophyll-a concentration. The additional example of the technique is intended for illustrative purposes, but should also serve as inspiration for similar applications. We dedicated a short subsection (7.3 *Performance analyses on remote sensing data*) to this additional example.

**Major comments Widespread use of unclear notation**

**RC2 – 4** One of the biggest issues with this paper is the widespread use of jargon and notation that is not common in the geosciences. For example, the paper makes widespread use of set notation (e.g., line 75, Equation 2), which is not commonly taught in geoscience curricula (I have no idea what the *f:* R × R... notation in Equation 2 means, and I have quite a bit of training in math and statistics, including having published in statistics journals myself!).

**RC2 – A4** We changed the function notation to:

$$f(x; h) = 1/nh \, \Sigma K((x-X_j)/h), \text{ where } x \epsilon R, h \epsilon R_{>0} \text{ and } f(x; h) \epsilon R_{\geq 0}$$

and changed the following part to include a definition of the involved sets: "The sets $R_{>0}$ and $R_{\geq 0}$ denote the positive real numbers and the non-negative real numbers, respectively. The *kernel function* satisfies..."

Furthermore, we changed the function notation in Eq. 4 to

$$\text{MISE}(\hat{f})(h) = E(\int R(\, \hat{f}\,(x; t) - f\,(x))^2 dx), \text{ where } h \, \Box \, R_{>0}$$

and Eq. 9 to

$$K_E\,(w) = \tfrac{3}{4}\,(1 - w^2)\,, \text{ where } w \,\epsilon\, R \text{ and } K_E\,(w) \,\epsilon\, R_{\geq 0}$$

and Eq. 10 to

$$\Phi\,(w) = 1/\sqrt{2\pi}\ e^{-\,1/2\,w2}, \text{ where } w \,\epsilon\, R \text{ and } \Phi\,(w) \,\epsilon\, R_{\geq 0}.$$

and Eq. 11 to

$$, \text{ where } x \,\epsilon\, R, t \,\epsilon\, R_{>0} \text{ and } \Phi\,(x; t) \,\epsilon\, R_{\geq 0}$$

and Eq. 36 to

$$\Phi_h\,(x) = \dots\,, \text{ where } x, \Phi_h\,(x) \in R$$

**RC2 – 5** In another example, := notation is used (e.g., line 225), and I am not sure what it means here; I'm used to reading it as `is distributed as', but I don't think that's what is meant here

**RC2 – A5** The "a:=b" notation means that "a is defined as b". We omitted this notation here, since it is already described in words next to the equation.

**RC2 – 6** (and what does the = : mean in equation 30?).

**RC2 – A6** Also here (and for the neighboring equations), we have omitted the notation and added describing sentences as:

"The nominator is approximated by the unbiased estimator and denoted as $E_\sigma \in R$.…" in ll. 288 – 289.

and: "... and set to $q_i \in R$ for all $i \in \{1, ..., n\}$ …." in l. 291.

and: "For the boundary values we set the second derivative at the lower boundary to $q_0 \in R$…" in l. 294.

and: "...and the second derivative at the upper boundary to $q_{n+1} \in R$…" in l. 296.

RC2 – 7 If notation like this is to be used, I suggest defining what the notation means at first use. My main concern here is that the notation may end up being a barrier to people reading this paper, which would then also inhibit this paper's potential impact.

**RC2 – A7** We checked our manuscript for further use of this notation.

We omitted the notation in the description of the spatial discretization: "...spatial discretization step size $R_{>0}$    $h = x_i - x_{i-1}$ for all $i \in \{1, \ldots, n\}$. For the following calculations, we set $x_{-1} = x_0 - h \in R$ and $x_{n+1} = x_n + h \in R$." in ll. 313 – 315.

and: "Now, we set ½  $1/h^2$  V $1/p$  = $A \in R^{(n+1)\times(n+1)}$, where $A \in R^{(n+1)\times(n+1)}$ means that A has real entries and n+1 rows and n+1 columns, the devision …" where we also added the clarification of the use of the set $R^{(n+1)\times(n+1)}$ in ll. 330 – 332.

We omitted the notation in the description of the temporal discretization in ll. 366 – 367.

And in Section 5.4 we omitted the notation in: "The boundary values are $x_{min} = \min X \in R$ and $x_{max} = \max X \in R$ by default, …" in ll. 431 – 432.

and changed the introduction of $A_{pilot}$ to: "reduces to a matrix denoted as $A_{pilot}$
½ $1/h^2$ V = $A_{pilot} \in R^{(n+1)\times(n+1)}$,
where $A_{pilot} \in R^{(n+1)\times(n+1)}$ means that $A_{pilot}$ has real entries and n+1 rows and n+1 columns." in ll. 450 – 452.

RC2 – 8 Somewhat related to the above, there is ambiguous usage of the symbol *t*. In some places it is clear that it is meant as the bandwidth (e.g., line 113). In others (e.g., the left-hand side of Equation 10), it seems to mean 'time' in the sense of time-evolution of the diffused quantity *u*. I don't think that the bandwidth and the time are meant to be taken as being equivalent in this paper, so a symbol other than *t* really should be used for the bandwidth. (Especially for this audience, where *t* is almost always reserved to refer to time.)

**RC2 – A8** The time parameter of the diffusion equation t can be identified with the squared bandwidth $h^2$, which is the motivation for the definition of a diffusion KDE and links both approaches. Any time t in the solution of the diffusion equation corresponds to a squared bandwidth $h^2$ of a KDE calculated from the classical sum/function definition. The final iteration time in the solution of the diffusion equation T determines a fixed bandwidth at which the diffKDE is evaluated.

We clarified this, by describing in Section 2 only the general form of a KDE with a general bandwidth h and moving the connection of the time parameter to the squared bandwidth to Section 3. For this, we deleted the sentence: "In the following we will exclusively deal with the squared bandwidth ($h^2$) and therefore adapt a notation where some t is defined as $h^2 =:$ tϵR." and replaced in Section 2 the variable t with the variable h and deleted the mentions of "squared" in this context.

We also restructured Section 3 to first explain, how the diffusion KDE is the solution of the diffusion equation: "This KDE solves Eq. 11, the partial differential equation describing the diffusion heat process, starting from an initial value based on the input data $(X_j)^N_{j=1}$ and progresses forward in time to a final solution at a fixed time $T \in R_{>0}$.
$\partial/\partial t\, u\,(x;\,t) = \frac{1}{2}\, d^2/dx^2\, u\,(x;\,t)\,,\, x \in \Omega,\, t \in R_{>0}$
The input data are treated as the initial value $u\,(x,\,0)$ at the initial time $t_0 = 0$ and generally set to infinitely high peaks at every data point $X_j$, $j \in \{1, ..., N\}$. The time propagation in solving Eq. 11 smooths the initial shape of u meaning that u contains less details of the input data $(X_j)^N_{j=1}$ for increasing values in time $t \in R_{>0}$. If we observe the solution u of Eq. 11 at a specific fixed final iteration time $T \in R_{>0}$, this parameter determines the smoothness of the function u and how many details of the input data are resolved. This is an equivalent dependency as already seen for the KDE as the solution of Eq. 2 depending on a bandwidth parameter $h \in R_{>0}$." in ll. 170 – 179.

and then how this KDE approach is linked to the Gaussian KDE by adding to the following paragraph: "This function solves Eq. 11 as the Green's function, where the time parameter $t \in R_{>0}$ equals the squared bandwidth parameter $h^2$ (Chaudhuri and Marron, 2000). Consequently, we can use the result of the optimal bandwidth from Eq. 13, only as the squared result as

$$T = ( \|K\|^2_{L2} / N\, k_2^2\, \|f''\|^2_{L2})^{2/5}$$

where we denote the optimal bandwidth now with $T \in R_{>0}$ as this is the final iteration time in the solution of Eq. 11." in ll. 184 – 189.

**RC2 – 9** That said, I'm genuinely confused about Algorithm 1, where lines 4 and 5 of the algorithm seem to clearly be setting $T_p$ and $T_f$ as bandwidths, but line 7 seems to be treating *t* as time and $T_p$ as a maximum time; so maybe bandwidth and time really are the same in this paper? If so, that needs to be stated *very* clearly and perhaps repeatedly, since that's quite unintuitive.

**RC2 – A9** We changed the reference to T, $T_p$ and $T_f$ in this section to "final iteration time" and deleted the reference: "...derived by the respective bandwidths."

Furthermore, we changed the reference to T, $T_p$ and $T_f$ in Section 4 to "final iteration time"

and rephrased l. 271 to: "We use a simple bandwidth as variants of the rule of thumb by Silverman (1986) to calculate both of them."

In the end, I'm not certain I understood the method well enough to review it thoroughly, which is a major concern.

**RC2 – 10** The introduction does a good job of describing KDE and the background of KDE, giving references to bandwidth choice up through Scott, 2012. However, there are some recent innovations in kernel density estimation that are highly relevant here:

Bernacchia and Pigolloti (2011) derive a method for choosing both the kernel and its bandwidth in an `optimal' way. Note also that there are several geoscience applications of this approach, which can be found by looking at literature citing this paper.
Davies and Baddeley (2017) describe a spatially adaptive bandwidth approach, which seems relevant here given the spatially-adaptive diffusivity used in this paper

Chacón and Duong (2018) overview KDE methods and the variety of bandwidth selection methods used; this is especially relevant since Duong authored the widely used ks package in R, which would be a good, additional state-of-the-science package to compare against in Figures 6, 7, 9.
Another relatively recent book overviews KDE methods:
https://link.springer.com/book/10.1007/978-3- 319-71688-6

I would also add that, for this audience, the introduction would be well-served by listing a number of examples of the use of KDE in geoscience literature.

**RC2 – A10** We have been in contact with Alberto Bernacchia after the publication of our preprint at GMD. Their approach differs from ours and reformulates the optimization of the KDE to an optimization of the shape of the kernel function for general bandwidths. Davies and Baddeley (2017) used an adaptive choice of the smoothing parameter depending on the spatial domain and applied an edge correction to achieve consistency at the boundaries, which is not necessary in our approach due to the inclusion of Neumann boundary conditions. We have added this different perspective to our introduction as: "… , to use adaptive bandwidth approaches (Davies and Baddeley, 2017) or to optimize the kernel function shape instead of the bandwidth (Bernacchia and Pigolotti, 2011)." in ll. 75 – 76.

Furthermore, we added Chacón and Duong (2018) and Gramacki (2018) to the reference list for bandwidth selection literature in l. 73 (not visibly resolved in the tracked changes document).

The ks package in R uses Gaussian kernels and provides as default the plug-in bandwidth selection, which we included for comparison already as the "GKDE" in the respective figures (calculated from the Python stats package).

Finally, we added the references for KDE applications in geosciences

O'Brien, J. P., O'Brien, T. A., Patricola, C. M., and Wang, S.-Y. S.: Metrics for understanding large-scale controls of multivariate temperature and precipitation variability, Climate Dynamics, 53, 3805–3823, https://doi.org/10.1007/s00382-019-04749-6, 2019.

Teshome, A. and Zhang, J.: Increase of Extreme Drought over Ethiopia under Climate Warming, Advances in Meteorology, 2019, 1–18, https://doi.org/10.1155/2019/5235429, 2019.

Ongoma, V., Chen, H., Gao, C., and Sagero, P. O.: Variability of temperature properties over Kenya based on observed and reanalyzed750

datasets, Theoretical and Applied Climatology, 133, 1175–1190, https://doi.org/10.1007/s00704-017-2246-y, 2017.

to the introduction by: "Well approximated PDFs have been used to benefit data analysis in geosciences (e.g. O'Brien et al., 2019; Teshome and Zhang, 2019; Ongoma et al., 2017)." in ll. 45 – 46.

**RC2 – 11** My final concern relates to the scope. In my initial thinking, I considered recommending rejection of this paper because it seems like it might be out of scope for GMD. Most of the content and innovation in the paper relates to a general-purpose statistical method, so it almost seems like this might be more suitable for a statistics journal like JABES.

**RC2 – A11** We revised many of the mathematical notations and phrases to make them more accessible (see RC2 – A1, RC2 – A4, RC2 – A5, RC2 – A6, RC2 – A7, RC1 – A12, RC1 – A18, RC1 – A20, RC1 – A27, RC1 – A33, RC1 – A34, RC1 – A35, RC1 – A36, RC1 – A37) and named the benefits of the diffusion KDE for the evaluation of geoscientific data clearly (see RC1 – A22) and repeated them in abstract, introduction and conclusion (see RC2 – A17).

Furthermore, we rephrased the initial paragraph of Section 7.4 to: "In geoscientific research, the derivation and comparison of well resolved PDFs can be useful, as demonstrated in our selected examples. Yet, the significance of resolving details in nonparamteric PDFs remains unclear. However, having high resolution PDFs available, as obtained with the diffKDE, is readily of value, and will likely guide further research. An obvious benefit of the diffKDE is its lesser dependence on the specification of a single, albeit optimal, bandwidth. Its application is likely more robust for the assessment of simulation results, either against data or results of other models (e.g.multi model ensembles), which is particularly relevant for evaluations of future climate projections obtained with Earth system models (e.g., Oliver et al., 2022).The presented diffKDE provides a nonparametric approach to estimate PDFs with typical features of geoscientific data. Being able to resolve typical patterns such as multiple or boundary close modes, while being insensitive to noise and individual outliers makes the diffKDE a suitable tool for future work in the calibration and optimization of Earth system models."

**RC2 – 12** I think that part of this impression comes across in the emphasis on the development of the statistical method itself, rather than its application (and/or potential application) in geosciences.

**RC2 – A12** In agreement with RC1, we split the results section up into two main sections showing the results on artificial and real geoscientific data individually to highlight the improvements made on each of them (see RC1 – A4).

**RC2 – 13** I think this partly relates to the technical notation comment above too; because the notation does not seem typical of a geoscience paper, it kind of feels like it was written for a stats audience.

**RC2 – A13** We revised many of the notations to be more accessible (see list provided in RC2 – A11).

**RC2 – 14** I spoke with GMD editors about this, and they indicated that they do think it is potentially in scope. But the paper should make a clear case for how this new method advances geoscientific modeling, and I think it might be good to emphasize applications a bit more than the method itself.

**RC2 – A14** We originally thought that the examples with the isotopic data and plankton size spectra would be illustrative enough. There are several ways of illustrating the performance of the diffKDE, and the value of estimation of nonparametric probability densities in general. We thought about this and have decided to provide an additional example of the application of the method (described in 7.3 *Performance analyses on remote sensing data*). The example should be simple and clear, as well as an inspiration for future potential applications. Our example shows detailed probability densities that cannot be easily represented by a parametric approach. It addresses temporal changes of spatial patterns within a specified region of interest, which we believe translates well to other applications in the field of geoscience.

**RC2 – 15** I also think that the discussion of the method (i.e., section 2) could be revised to be much less rigorous (which would also help with the statistical notation comment); focus on making sure that a large portion of your potential geoscience audience can understand this method (and your innovation to it) rather than focusing on precise mathematical language.

**RC2 – A15** We conducted a major revision of Section 2, completely restructuring it (see RC1 – A1, RC1 – A2, RC1 – A3) and simplification and clarification of the statistical details (see RC2 – A11).

**RC2 – 16** It also might help to modify the introduction by mentioning some specific uses of KDE in geoscience papers and also modifying the discussion to relate the advances in this paper back to those papers; what might have been different/better about the past papers if they had used your method?

**RC2 – A16** We agree and follow the reviewers suggestion. The first paragraphs of the introduction have been revised. We now refer to scientific papers that put emphasis on the use of nonparametric PDFs. However, an evaluation of the quality of the estimated PDFs published in previous studies is not really possible and might not be useful. Even if we could do so, the refined PDF estimates need not automatically alter the scientific inferences documented in these publications. This would go beyond the scope of our method description presented here. The metrics that are applied to quantify the similarity between the PDFs could be more critical, which we also mention.

As far as advances are concerned, there is a clear exception. It is the application of the KDE for the derivation of size spectra, which is the reason we have included such an example. We pick up this topic now already in the introduction, which should make it straightforward for readers to see why we have worked out an example in this regard. See also RC2 – A10.

**RC2 – 17** I would also add that the paper should be more explicit about the innovations of this specific paper: in the abstract, the introduction, in the method section, and again in the conclusions.

**RC2 – A17** In agreement with RC1, we highlighted the three main advantages of the diffusion KDE in Section 3 as: "This different calculation offers three main advantages: (1) consistency at the boundaries can be ensured by adding Neumann boundary conditions (2) better resolution of multimodal data can be achieved by the inclusion of a suitable parameter function in the differential equation leading to adaptive smoothing intensity (3) a family of KDEs with different bandwidths is produced as a by-product of the numerical solution of the partial differential equation in individual time steps." in ll. 165 – 169.

We included these benefits in the abstract as: "In this study, we designed and developed a new implementation of a diffusion-based KDE as an open source Python tool to make diffusion-based KDE accessible for general use. Our new diffusion-based KDE provides (1) consistency at the boundaries, (2) better resolution of multimodal data, (3) and a family of KDEs with different smoothing intensities." in ll. 8 – 11.

and in the introduction: "This different approach comes with three main advantages: (1) consistency at the boundaries (2) better resolution of multimodal data (3) a family of KDEs with different smoothing intensities can be produced as a by-product of the numerical solution." in ll. 78 – 80.

and in the conclusion as: "We chose this approach to KDE, because it offers three main benefits: (1) consistency at the boundaries (2) better resolution of multimodal data (3) a family of KDEs with different smoothing intensities. We provide our algorithm in an open source Python package." in ll. 760 – 762.

**RC2 – 18** I do see that the innovation of this paper is discussed in lines 544-549, which is great, but the way this is written, it is not clear what specific aspects of this paper are new relative to the citations mentioned.

**RC2 – A18** We rephrased this part to: "… Our approach includes a new approximation of the bandwidth, which equals the square root of the final iteration time. We directly approximate the analytical solution of the optimal bandwidth with two pilot estimation steps and finite differences. We calculate the pilot estimates as solutions of a simplified diffusion equation up until final iteration times derived from literature based bandwidths called *rule of thumb* by Silverman (1986). Our new approach results in three subsequent estimations of the PDF, each of them chosen with a finer bandwidth approximation." in ll. 762 – 768.

Furthermore, we moved the description of our new parts of our method (novelties) to Section 4, becoming a section of its own. And the implementation in Python is described in Section 5.

**RC2 – 19** It might help to revise Section 2 such that it is clear which equations are essentially 'background information' and which equations (or which parts of the equations) contain the innovation in this paper.

**RC2 – A19** In agreement with RC1, we restructured Section 2 completely, now we distinguish between background information with regard to i) the general KDE (Section 2), ii) the diffusion KDE (Section 3), and iii) our new approach (Section 4), and iv) the specific

discretization discretization and implementation of our diffKDE (Section 5) (see RC2 – A15).

[revised manuscript text omitted]
}\left(p\left(X\right)\right) = \frac{1}{n+1}\sum_{i=0}^{n+1}\sqrt{p\left(x_i\right)} =: E_\sigma$$

and the second derivative in the denominator by finite differences (McSwiggan et al., 2016)

$$\left(\frac{f}{p}\right)''\left(x_i\right) = \frac{1}{h^2}\left(\frac{f}{p}\left(x_{i+1}\right) - 2\frac{f}{p}\left(x_i\right) + \frac{f}{p}\left(x_{i-1}\right)\right) =: q_i$$

for all $i \in \{1,...,n\}$. For the boundary values we set

$$\left(\frac{f}{p}\right)''\left(x_0\right) = \frac{1}{h^2}\left(2\frac{f}{p}\left(x_1\right) - 2\frac{f}{p}\left(x_0\right)\right) =: q_0$$

and

$$\left(\frac{f}{p}\right)''\left(x_{n+1}\right) = \frac{1}{h^2}\left(2\frac{f}{p}\left(x_{n-1}\right) - 2\frac{f}{p}\left(x_n\right)\right) =: q_{n+1}.$$

[revised manuscript text omitted]